# PINN Balls: Scaling Second-Order Methods for PINNs with Domain Decomposition and Adaptive Sampling

**Andrea Bonfanti**
BMW AG, Digital Campus Munich
Basque Center for Applied Mathematics
University of the Basque Country
abonfanti001@ikasle.ehu.eus

**Ismael Medina**
University of Göttingen,
Campus Institute Data Science

**Roman List**
BMW AG, Digital Campus Munich

**Björn Staeves**
BMW AG, Digital Campus Munich

**Roberto Santana**
University of the Basque Country,
Intelligent Systems Group
roberto.santana@ehu.eus

**Marco Ellero**
Basque Center for Applied Mathematics, CFD Group
IKERBASQUE, Basque Foundation for Science
Swansea University, Complex Fluids Research Group
mellero@bcamath.org

## Abstract

Recent advances in Scientific Machine Learning have shown that second-order methods can enhance the training of Physics-Informed Neural Networks (PINNs), making them a suitable alternative to traditional numerical methods for Partial Differential Equations (PDEs). However, second-order methods induce large memory requirements, making them scale poorly with the model size. In this paper, we define a local Mixture of Experts (MoE) combining the parameter-efficiency of ensemble models and sparse coding to enable the use of second-order training. Our model – PINN BALLS – also features a fully learnable domain decomposition structure, achieved through the use of Adversarial Adaptive Sampling (AAS), which adapts the DD to the PDE and its domain. PINN BALLS achieves better accuracy than the state-of-the-art in scientific machine learning, while maintaining invaluable scalability properties and drawing from a sound theoretical background.

## 1 Introduction

Obtaining accurate solutions to partial differential equations (PDEs) is a ubiquitous problem in countless engineering applications, as it enables reliable simulations of real-world scenarios. While discretization-based methods for solving PDEs offer convergence guarantees as well as a solid theoretical underpinning, their implementation and runtime can be resource intensive. Novel techniques developed within the domain of Scientific Machine Learning (SciML) strike a highly sought-after balance between accuracy and runtime, which has led to a rapid growth of the field in recent years [7]. A milestone SciML architecture is the Physics-Informed Neural Network (PINN), which was theorized in the 90s [9] and gained the interest of the research community following the publication that coined its name [28]. The salient aspect of PINNs is the inclusion of physical laws in the loss function, computed via automatic differentiation. While their training process is not simple, researchers proposed alternatives to fix pertinent issues of the PINN architecture, such as the complex loss landscape generated by the PDE residuals [36], unbalanced optimization of the components of the loss functions [17], and the spectral bias intrinsic to neural networks [29]. The combination of

39th Conference on Neural Information Processing Systems (NeurIPS 2025).

those with second-order optimizers has proven to be an invaluable approach to surpass the limitations of PINNs [4, 24]. Second-order methods are a crucial step to enable highly accurate solutions, which can be a strict requirement in some applications. However, these methods scale poorly with an increasing number of parameters.

Scaling to large number of parameters is a crucial component for PINN architectures as the majority of problems simulated in industrial scenarios typically feature highly complex solutions and arbitrarily large domains. The majority of approaches in the literature rely on domain decomposition (DD), which is a common approach adopted by traditional PDE solvers. DD divides the task of predicting the PDE solution into different subdomains, which are routed to different submodels. In the machine learning community, architectures that combine multiple models are regarded as ensemble models. A widely known example of an ensemble model is the Mixture of Expert (MoE) [22], which approximates a function as a linear combination of submodels (or experts) driven by a – possibly learnable – weighting factor, which is referred to as a "gating mechanism". This allows the MoE to scale well to larger problems and often to parallelize the training of the submodels [32], strongly limiting the computational overhead when several models are combined [31]. While a natural choice for scalability in the field of PDEs, the use of DD involves several challenges. A major issue appears when equations contain non-local quantities – eg. the pressure in incompressible Navier-Stokes equations – that cannot be easily inferred from local information. This approach is therefore typically complemented by the addition of a loss component at the interface of neighboring models. However, these interface conditions can strongly hinder the training of the model and force the predicted solution to oscillate between local minima. Furthermore, there are several ways of defining interface conditions for ensemble models; the choice is typically problem-dependent and can not be defined a priori [21].

An additional complexity introduced by DD is intrinsic to the decomposition itself. Since real-world scenarios typically include multi-scale and finely detailed domains, it is imperative to be able to adapt the decomposition to the underlying domain and PDE. Indeed, problems in the field of fluid dynamics are typically inhomogeneous and they require adaptive meshes for both stability and scalability. Therefore, the DD can not always be decided a priori. It has to be learned alongside the PDE solution, and possibly be adapted both to the geometry of the problem and to the nature of the PDE that is being solved. However, most of the literature works combining DD and PINNs do not instantiate such a domain decomposition. In most cases, the DD is not trainable but rather fixed beforehand. To address this issue we adopt the Adversarial Adaptive Sampling framework (AAS) proposed in [33], which provides a solid theoretical framework for dynamic sampling that supports the training process and guides the DD.

**Related Work**    The synergy between PINNs and DD has already been explored for example by Extended-PINNs (XPINN) [16], which relies on non-overlapping subdomains and interface conditions to match the prediction of neighboring submodels. Another well-known example is the Finite-Basis PINN (FBPINN) [23], which adopts predefined overlapping sub-domains and leverages the sparsity of the model's output to compute parameter updates without relying on interface conditions. This model has been extended to Extreme Learning Machines [2] with the goal of achieving highly accurate solutions. A few additional works also propose to include a trainable domain decomposition, such as the Augmented PINN [14], which represents an extension of the XPINN to a soft pre-trained domain decomposition. Similarly, the MoE PINN [3], which takes advantage of the ensemble structure and the agreement of the submodels as a natural approach for uncertainty quantification. We refer to [13] for a non-exhaustive overview of existing methods.

**Motivation and Goal**    In our research, we propose to combine the benefits of DD, second-order training methods and Adversarial Adaptive Sampling (AAS). Indeed, the paradigm of ensemble models for DD synergizes with the use of second-order methods: while the resource efficiency of second-order methods scales poorly with the number of parameters, ensemble models are parameter-efficient and their structure allows for parallel evaluation of the model and its gradients [23]. Moreover, we propose to base the DD on overlapping radial basis functions to emphasize locality, hence restricting the prediction of each submodel to a ball within the PDE domain. This approach ensures that the Jacobian matrix required for computing the update step of the second-order method is sparse, relieving the memory-intensive nature of second-order optimizers as Natural Gradient Descents [24] and Levenberg-Marquardt [4]. Last but not least, we leverage the theoretical backbone of AAS [33] to train the DD alongside the ensemble model, which requires no a priori knowledge of the features

Table 1: Difference in features between PINN BALLS, the APINN [14] and the FBPINN [23].

| | APINN | FBPINN | PINN BALLS |
|---|---|---|---|
| Sparse | | ✓ | ✓ |
| No interface loss | | ✓ | ✓ |
| Trainable DD | ✓ | | ✓ |
| Adaptive Sampling | | | ✓ |
| Second-order training | | | ✓ |

of the PDE solution or the computational domain. This allows the DD to direct its attention to PDE features that are harder to compute, enhancing the accuracy and robustness of the method.

The paper is structured as follows: Section 2 introduces the notation for PINNs and DD. Section 3 introduces AAS, showcasing our theoretical advancements in 3.1. The complete PINN BALLS model is presented in Section 4, and its scalability with respect to the number of parameters of the models is discussed in 4.1. The full training pipeline is then highlighted in Section 4.2, and its performance is evaluated in Section 5. Limitation are discussed in Section 5.3 and the paper is concluded with Section 6. Our contributions are gathered in Table 1. To summarize:

- To the best of our knowledge, we are the first to combine second-order training methods with domain decomposition on PINNs, including scaling to large parameters;
- We adopt AAS to introduce a highly adaptable framework to train the domain decomposition;
- We advance the mathematical foundation of AAS, ensuring robustness of the model.

Our numerical results show outstanding performance on benchmark PDEs, demonstrating a consistent behavior of the parameter efficiency and global accuracy with respect to the size of the ensemble model. This also holds for cases such as the Navier-Stokes equations, which involves particular challenges for DD due to the aforementioned non-local features.

## 2 Physics-Informed Neural Networks and Domain Decomposition

The PINN architecture leverages the power of neural networks as functional approximators to address classical PDE solution instances on a bounded domain $\Omega \subset \mathbb{R}^{d_{\text{in}}}$. Without loss of generality, consider the problem of finding a function $u : \Omega \to \mathbb{R}^{d_{\text{out}}}$ which satisfies the following equations:

$$\begin{aligned} \mathcal{R}u(x) &= f(x), & x \in \Omega, \\ u(x) &= g(x), & x \in \partial\Omega, \end{aligned} \tag{1}$$

where the underlying PDE is fully defined by the differential operator $\mathcal{R}$, and boundary and initial conditions are collected in a known function $g$. PINNs aim to approximate the solution of the aforementioned PDE through a neural network $u_\theta : \Omega \to \mathbb{R}^{d_{\text{out}}}$ with $L$ layers, defined as:

$$u(x; \theta) := W_{L-1} \cdot \sigma(\cdots \sigma(W_0 x + b_0) + \cdots) + b_{L-1}. \tag{2}$$

where $W_i \in \mathbb{R}^{h_i \times h_{i-1}}$ and $b_i \in \mathbb{R}^{h_i}$ denote respectively the weights matrix and bias vector of the $i$-th hidden layer with dimension $h_i$. For the sake of compactness, $\theta$ represents the collection of all the trainable parameters of the network, i.e. $\theta = \{W_i, b_i\}_{i=0}^{L-1}$. The activation function $\sigma : \mathbb{R} \to \mathbb{R}$ is a smooth coordinate-wise function, such as the hyperbolic tangent, or the sine function which are common choices for PINNs. The "Physics-Informed" nature of the neural network $u_\theta$ lies in the choice of the loss function chosen for minimization, which is typically given by

$$\mathcal{L}(\theta) := \int_\Omega |\mathcal{R}u(x; \theta) - f(x)|^2 dx + \int_{\partial\Omega} |u(x; \theta) - g(x)|^2 d\omega(x). \tag{3}$$

In the above formulation, $dx$ and $d\omega$ represent infinitesimal volume and surface element respectively. The integrals in (3) are typically approximated as:

$$L(\theta) := \frac{1}{|N_\Omega|} \sum_{i=1}^{N_\Omega} |\mathcal{R}u(x_i; \theta) - f(x_i)|^2 + \frac{1}{|N_{\partial\Omega}|} \sum_{k=1}^{N_{\partial\Omega}} |u(\hat{x}_k; \theta) - g(\hat{x}_k)|^2, \tag{4}$$

where $N_\Omega$ represents the number of collocation points $\{x_i\}_{i=1}^{N_\Omega}$ in $\Omega$ to minimize the PDE residuals and $N_{\partial\Omega}$ the number of points $\{\hat{x}_k\}_{k=1}^{N_{\partial\Omega}}$ used to fit the initial and/or boundary conditions at $\partial\Omega$.

## 2.1 Domain Decomposition

The concept of domain decomposition is ubiquitous in the field of numerical solution of PDEs and it has been seamlessly used in order to tackle the scalability of PINNs. The rationale is to divide the PDE domain $\Omega$ into $M$ — possibly overlapping — subdomains $\{\Omega_j\}_{j=1}^M$, such that $\bigcup_{j=1}^M \Omega_j = \Omega$. Each subdomain can be associated with a local solution or, in the case of PINNs, with a local neural network $U_j(x) := U(x, \theta_j)$ parameterized by a subset of $\theta$ denoted by $\theta_j$. The aforementioned splitting allows parallelizable training routines, which are beneficial in terms of scalability [31] with respect to the number of parameters and domain size, and in the effectiveness and accuracy of the model [10]. Therefore, the overall model $u_\theta$ is obtained as a combination of the local models:

$$u_\theta(x) = \sum_{j=1}^M \lambda_j(x)U_j(x) \quad \text{s.t.} \quad \sum_{j=1}^M \lambda_j(x) = 1, \ \lambda_j(x) \geq 0 \quad \forall j = 1, ..., M, \quad \forall x \in \Omega. \quad (5)$$

Here, $\lambda(x) := (\lambda_1(x), \ldots, \lambda_M(x))$ may be interpreted as a vector of weighting factors indicating which submodel has more impact in the prediction of $u_\theta$ at $x$, and is commonly referred to as the *gating function*. The latter property of $\lambda$ is typically enforced to impose a proper partition of unity.

The most popular works on domain decomposition in PINNs [16, 19] also rely on the introduction of interface conditions to align the prediction of submodels between subdomains. Choosing the correct interface condition is a nontrivial, domain-dependent task [21]. Typical interface conditions require to evaluate the difference between each pair of overlapping bases $U_j$ and $U_i$. While this step can be computed efficiently through parallelization, interface conditions require an additional loss component. As the nuanced interplay between loss components is known to introduce unfavorable complexity in the training of PINNs [20], we advocate for a global optimization step, which can still be computed efficiently due to the locality of each submodel [23].

## 3 Adversarial Adaptive Sampling

Adversarial Adaptive Sampling was introduced in [33] as a technique to enhance the training of PINNs. Its core idea is to sample the collocation points from a probability density $p \in \mathcal{P}(\Omega)$ to be learned simultaneously with the PINN, following the blueprint of generative adversarial neural networks. This corresponds to the minimax problem:

$$\min_\theta \max_{p_\phi \in \mathcal{P}(\Omega)} \mathcal{J}(\theta, p_\phi), \qquad \text{with } \mathcal{J}(\theta, p_\phi) := \int_\Omega |\mathcal{R}u(x; \theta) - f(x)|^2 p_\phi(x)dx - \beta G(p_\phi), \quad (6)$$

where $p_\phi$ is a parameterized version of $p$ and $G$ is a regularizing term that prevents the collapse of $p_\phi$ to a Dirac delta. The intuition is that $p_\phi$ should emphasize regions where the residuals are difficult to minimize, while ensuring full coverage of the PDE domain. The founding paper [33] shows existence of solutions for the minimax problem (6) with $G$ chosen to constrain the Lipschitz constant of $p_\phi$. Nevertheless, [33] employs in the numerical examples the more practical regularization $\int_\Omega |\nabla p_\phi(x)|^2 dx$, for which a convergence proof is not provided. We bridge this gap in Section 3.1.

### 3.1 Convergence Result

To lay down a sound theoretical framework, we extend the theoretical results in [33] to practical regularization functionals for the probability distribution $p$. In particular we complete the rigorous treatment of the regularization term $\int_\Omega |\nabla p(x)|^2 dx$ proposed by [33], as well as an entropic regularization term motivated by the field of optimal transport [8]:

**Theorem 3.1.** *Assume one of the following settings:*

1. *$G$ is the Dirichlet energy $G_D(p) := \int_\Omega |\nabla p(x)|^2 dx$, and $\inf_\theta \|(\mathcal{R}u_\theta - f)^2\|_2 = 0$, or*

2. *$G$ is the negative entropy $G_E(p) := \int_\Omega p(x) \log p(x)$, and $\inf_\theta \|(\mathcal{R}u_\theta - f)^2\|_{C(\Omega)} = 0$.*

*Then the optimal value of the minimax problem (6) is 0. In particular, for any sequence $\{\theta_n\}_n$ converging to the infimum assumed above, there exists a sequence of probability measures $\{p_n\}_n$ (with $p_n$ maximizing $\mathcal{J}(\theta_n, \cdot)$ such that*

$$\lim_{n \to \infty} \mathcal{J}(\theta_n, p_n) = 0. \quad (7)$$

*Moreover, the probability density functions $\{p_n\}_n$ converge to the uniform distribution on $\Omega$.*

*Idea of the proof.* The uniform density $\tilde{p} = 1/|\Omega|$ constitutes a lower bound for the nested maximization step in (6). This fact implies a certain coercivity of the maximization problem, which can be used to show existence of a maximizer $p^*$; combining this aspect with a minimizing sequence for the residuals completes the proof. The detailed proofs can be consulted in Sections A.1 and A.2;  □

The entropic regularization $G_E$ proposed in Theorem 3.1 and the gradient-based loss $G_D$ of [33] share its aim to smooth out $p$. In addition, the entropic regularization $G_E$ features an infinite slope at the origin that inhibits the probability density from reaching zero anywhere within the domain. Hence, positivity of $p_\phi$ is baked into $G_E$, while it needs to be additionally imposed for $G_D$.

*Remark* 3.2. As shown by [18, 26], both the gradient flow of the Dirichlet energy with respect to the $L^2$-topology and the gradient flow of the entropy with respect to the $L^2$-Wasserstein distance generate the same dynamics: the heat equation. This suggests a compelling interpretation: a gradient ascent step in (6) may be seen as the application of a heat kernel that smooths out $p$ with respect to the landscape of the residuals. For more context and implications of this intuition see Section A.3.

*Remark* 3.3. The novelty of Theorem 3.1 is twofold: on the one hand we extend the convergence results of [33] to regularization functions that are more practical to implement, since the Lipschitz constraint considered in [33] is hard to implement in practice. On the other hand we show the convergence of $p_n$ to a uniform distribution along any converging subsequence, which in particular implies that the sequence of maximizing distributions $\{p_n\}_n$ tends to cover the full domain as the residual of $u_n$ converges to zero. This result further validates the robustness of our method, ensuring an adequate generalization to arbitrarily complex PDE domains.

## 4 Sparse Ensemble of Physics-Informed Neural Networks

A crucial step in the proposed model is the usage of Radial Basis Functions (RBFs) to identify the domain decomposition. In particular, we opt for quartic basis functions as in Equation 8 – instead of the more traditional Gaussian RBF – to emphasize locality of the submodels.

$$\lambda_j(x) = \frac{\phi_j(x)}{\sum_{l=1}^{M} \phi_l(x)} \qquad \text{where} \qquad \phi_j(x) = \begin{cases} \left(1 - \frac{|x-c_j|^2}{s_j^2}\right)^2 & \text{if} \quad \frac{|x-c_j|}{s_j} \leq 1 \\ 0 & \text{elsewhere} \end{cases} \tag{8}$$

The centers and radii of each ball $\phi_j$ are indicated respectively as $c_j$ and $s_j$, and identifies the trainable parameters of the domain decomposition. We adopt the notation $\Omega_j$ to indicate the compact support of the basis function $\phi_j$, i.e. the set $\{x \in \Omega \quad \text{s.t.} \quad \phi_j(x) > 0\}$. The final gating function $\lambda_j$ is obtained by normalizing $\phi_j$ with respect to the total sum of RBFs. This ensures a proper partition of unity while maintaining the sets $\Omega_j$ compact and small. Ensuring that $\sum_{l=1}^{M} \phi_l(x) > 0$ for all $x$ can be done by choosing sufficiently large $\{s_j\}_{j=1}^{M}$ at initialization.

**Connecting Domain Decomposition and Adversarial Adaptive Sampling** In order to align the training of the parameters of each $\phi_j$ with the training of the PINN parameters, we devise a connection between the domain decomposition and the sampling distribution $p_\phi$. In particular, we opt to approximate $p_\phi$ through the combination of the basis $\phi_j$ as:

$$p_\phi(x) = \frac{1}{M} \sum_{k=1}^{M} w_k \phi_k(x), \tag{9}$$

where $w_k$ is chosen so that $w_k \phi_k$ is a probability distribution. This parametrization of $p_\phi$ enforces the domain decomposition to focus on areas where the PDE residuals are higher, which typically occurs in regions of the domain where the PDE solution presents steep gradients and/or high frequency components. Moreover, it implicitly defines a suitable training proceedure for the DD, given by the gradient ascent step of the minimax training routine.

**Definition 4.1** (PINN BALLS). We define PINN BALLS a tuple of the form $(u_\theta, p_\phi)$, where $u_\theta$ is the ensemble model defined in (5), with a domain decomposition structure $\{\lambda_j\}_{j=1}^{M}$ given by (8), and $p_\phi$ is the probability distribution of the form (9). The loss function is given by discretizing the AAS loss (6) on a set of points $\{x_i^\phi\}$ sampled from $p_\phi$:

$$J(\theta,\phi) := \frac{1}{N_\Omega} \sum_{i=1}^{N_\Omega} |\mathcal{R}u(x_i^\phi; \theta) - f(x_i^\phi)|^2 + \frac{1}{N_{\partial\Omega}} \sum_{k=1}^{N_{\partial\Omega}} |u(\hat{x}_k) - g(\hat{x}_k)|^2 - \frac{\beta}{N_\Omega} \sum_{i=1}^{N_\Omega} \log p_\phi(x_i^\phi). \quad (10)$$

*Remark* 4.2. Ensemble models of the form 5 are universal approximators. This holds trivially as each submodel is a universal approximator itself.

## 4.1 Scalable second-order training of the parameters

To train our model we adopt the Levenberg Marquardt update rule, which is a second-order quasi-Newton method that represents a stabilized version of the Gauss-Newton method [25].

$$\theta^{k+1} = \theta^k - (D_\theta L^T D_\theta L + \eta I)^\dagger \nabla_\theta L(\theta^k). \quad (11)$$

Here $D_\theta L$ denotes the Jacobian of $L$ with respect to the model parameters, and the superscripts $k$ and $\dagger$ represent respectively the training iteration and the Moore-Penrose pseudo-inverse. The scalar value $\eta$ is determined heuristically during training, balancing the interplay between a gradient descent step and a second-order update. The latter is necessary to efficiently reach a suitable minimum, whilst the former is useful in practice at early iterations.

The most prominent drawback of this training routine is given by the storage and inversion of the matrix $D_\theta L^T D_\theta L$ in GPU memory, which is highly impractical as its size scales quadratically with the number of parameters of the model. A practical solution to this is to ensure that the aforementioned matrix can be stored in a sparse format, indicating only a linear growth with respect to the number of parameters. This can be done by leveraging the locality of the submodels of PINN BALLS, since the effect of each $U_j$ is restricted to the compact support $\Omega_j$.

Indeed, the components of $D_\theta L$ are defined as the partial derivatives of $u_\theta$ with respect to the parameters $\theta$. Outside of the compact support $\Omega_j$, hence where $\lambda_j$ and all its derivatives are identically 0, the jacobian components of $U_j$ vanish. This implies that in the aforementioned region it is not necessary to backpropagate to obtain the partial derivatives necessary to compute the PDE residuals.

$$D_\theta L = \begin{bmatrix} \partial_\theta \sum_{j=1}^M \lambda_j U_j \\ \partial_\theta \mathcal{R} \left[ \sum_{j=1}^M \lambda_j U_j \right] \end{bmatrix} = \begin{bmatrix} \sum_{j=1}^M \lambda_j \partial_\theta U_j \\ \partial_\theta \mathcal{R} \left[ \sum_{j=1}^M \lambda_j U_j \right] \end{bmatrix}. \quad (12)$$

On top of sparsity, the PINN BALLS model also enables parallel implementations for the computation of $D_\theta L$. Sparse and parallel implementations combined allows to adopt the exact formulation of the LM update step by computing $D_\theta L^T D_\theta L$ in a fast and memory efficient fashion. The inversion remain the ultimate challenging aspect, which can be done through sparse solvers. Another valuable alternative is to consider matrix-free multiplication methods as conjugate gradient, whose implementation can benefit from the sparse nature of our Jacobian. This class of methods is however out of the scope of the present paper, since they are often sensitive to poor conditioning of the underlying matrix, which is often the case when training PINNs. Finding a viable preconditioning strategy remains thus an interesting direction of future study.

## 4.2 Min-max Training Routine

The training of the PINN BALLS model follows the alternating descent-ascent approach proposed for AAS in [33]. For the parameters $\theta$ of each submodel we adopt the Levenberg-Marquardt (LM) update step defined in (11). When the problem is relatively simple – e.g. linear PDEs – one can also resort to a more efficient approach of alternating a gradient descent step on $\theta$ and a second-order fine tuning of the parameters of the last layer of the model, which is closely connected to the approach of Extreme Learning Machines [15] and close to that employed in [2].

The training step for the domain decomposition $\phi$ consists in a gradient ascent step of (10). As proposed in [33], it is convenient to rewrite (10) using importance sampling:

$$F(p_\phi) := \frac{1}{N_\Omega} \sum_{i=1}^{N_\Omega} \frac{|\mathcal{R}u(x_i^\phi; \theta) - f(x_i^\phi)|^2 p_\phi(x_i^\phi)}{p_{\bar{\phi}}(x_i^\phi)} - \frac{\beta}{N_\Omega} \sum_{i=1}^{N_\Omega} \frac{p_\phi(x_i^\phi) \log p_\phi(x_i^\phi)}{p_{\bar{\phi}}(x_i^\phi)}. \quad (13)$$

Here $p_{\bar{\phi}}$ stands for the current value of $p_\phi$, which is taken to be a constant and therefore does not enter the computational graph. (13) is evaluated on the same sample points as the $\theta$-update, and the

boundary points are ignored, since they do not depend on $\phi$. The choice of the learning rate for the parameters of $p_\phi$ is important for stabilizing the overall training routine. In particular, we notice that the LM training shows better performance when $p_\phi$ does not strongly vary over training iterations. However, its fluctuations during training are important to ensure proper convergence of the model. Hence, we recommend an exponential decay for its learning rate to emphasize variability in early iterations and favor convergence of the model in late training stages. The complete training pipeline is summarized in Figure 1.

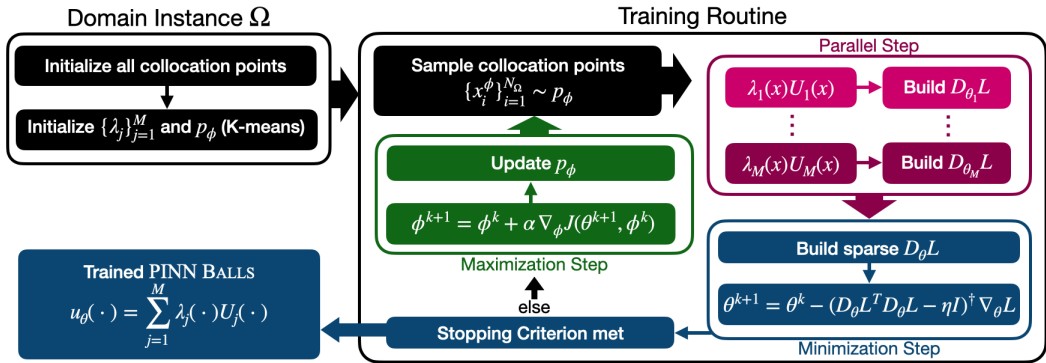

Figure 1: High-level diagram of the minimax training routine devised for the PINN BALLS model.

## 5 Numerical Evaluation

We begin by evaluating the behavior of the Levenberg–Marquardt (LM) update step on a simple supervised problem — approximating the function $f(x, y) = \sin 4x \sin y$. For various number of parameters, we instantiate a standard dense neural network and PINN BALLS architectures with 5, 10, 20, and 50 PINN Balls. The disposition of the centers $c_j$ is determined by K-means over the collocation points, and $s_j$ is selected to be the minimum possible to cover all training samples, which is identified by the standard deviation of each cluster. For each model, we perform a single LM update and collect statistics on runtime, Jacobian sparsity and memory consumption; results are summarized in Figure 2. In the case of PINN BALLS, we exploit the sparsity of the Jacobian by constructing it in Compressed Sparse Row (CSR) format using the `scipy.sparse` package, solving the resulting system either in the `scipy` package [35] via direct dense or sparse solvers depending on the structure. The statistics shown do not refer to the accuracy of the model nor include the computation of the matrix $D_\theta L^T D_\theta L$, respectively discussed in Sections 5.1 and 5.3.

**Efficiency of the matrix inversion** Although dense solvers generally achieve faster wall-clock times for moderate problem sizes, models employing a large number of PINN Balls achieve competitive runtimes. Notably, highly sparse models (50 PINN Balls) present comparable runtimes even in moderate problem size, which is mainly due to numerical heuristic of sparse matrices: visible performance improvements are typically noticeable when the nonzero elements of the matrices are 10% or less of the total.

**Memory Efficiency** From a memory perspective, using a large number of PINN Balls provides significant gains, which is to be expected due to the sparse nature of the Jacobians. However, the dense architecture is more memory efficient than the 5-PINN Balls model, whose Jacobian presents 30% of nonzero elements. Indeed, in the sparse formulation the nonzero values are stored alongside their indices, which results in roughly 3 times the number of nonzero values in the matrix. Notably, for the simple supervised problem considered, the dense solver reaches an out-of-memory error at roughly 8000 parameters., while the sparse ones can handle much larger models.

To assess the robustness of these findings, we replicated the experiment across a range of target functions of varying complexity. Remarkably, the trends observed in Figure 2 persist almost independently of the choice of $f$. While in theory the condition number of $D_\theta L$ could impact the difficulty of solving the LM update, we observe that the damping effect inherent to the LM formulation sufficiently regularizes the system, mitigating potential instabilities due to ill-conditioning. Note that PINNs

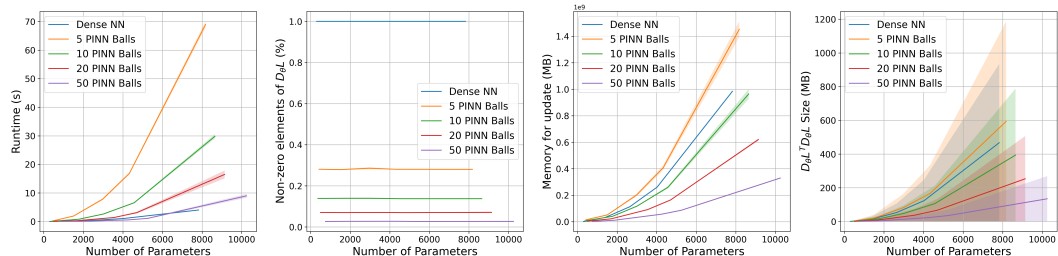

Figure 2: Mean value with shaded variance, for different ensemble sizes, of various statistics of the LM update step on PINN BALLS. From left to right, the runtime in seconds, the percentage of non-zero elements of $D_\theta L$, and the CPU memory consumed to invert and store the matrix $D_\theta L^T D_\theta L$

trained with second-order methods show very strong performance even with few parameters when compared to larger models trained with first-order methods or LBFGS [4]. Notably, the sparsity structure of the Jacobian may be altered during training by the DD, due to changes in the basis radii. However, when preserving high sparsity is critical, the basis radius can be capped or even fixed after initialization.

## 5.1 Training on Benchmark PDEs

We now showcase the results obtained by the PINN BALLS model on three well-known test cases for PINNs: The Helmholtz Equation, Burgers' Equation and time-dependent, incompressible Navier Stokes Equations in 2D. We show the relative $L^2$ error achieved during training as the number of submodels increases, keeping the same architecture for each individual submodel. These results are collected in Figure 3 and commented below. For the sake of brevity, we gather details of the PDEs solved and additional results in Appendix B. We do not include comparison with first order optimization. We refer to [4] for a thorough comparison between first and second order optimization in general. Combined with the DD settings, tendentially first-order optimization is slower, especially in a scenario like ours which includes an adversarial minimax formulation. All models are implemented in JAX [5] using double precision and trained on a single NVIDIA A10 GPU.

**Helmholtz Equation**  The leftmost plot of Figure 3 showcases the training loss achieved on the Helmholtz equation. Increasing the ensemble size also enhances the fluctuations in the loss function. This behavior is expected, as increasing the number of basis makes the maximization step in (13) more accurate. However, the very small relative $L^2$ error achieved shows that the PINN BALLS model is capable of obtaining extremely accurate solutions. Remarkably, the $L^2$ error with respect to the ground truth does not present high frequency components, which is a common artifact in PINN solutions. Moreover, the $L^2$ error in the order of $10^{-10}$, which is notably low for a PDE solver in general. The final distribution of the basis $\phi_j$ does not present any noteworthy pattern.

**Burgers' Equation**  The center plot of Figure 3 showcases the $L^2$ Error achieved during training on Burgers' equation. It is possible to notice that strong oscillation in accuracy occurs during early iterations, which is often due to the higher complexity that can be captured in the maximization step by $p_\phi$. This behavior proves to be beneficial for the model, as it allow to reach lower $L^2$ errors, up to the order of $10^{-4}$ for a model with 20 PINN Balls.

**Navier-Stokes Equations**  Navier Stokes equations represent a truly challenging scenario for PINNs, as the traditional implementation fails to converge on the classical benchmark of a fluid flow past a cylinder in 2D [6]. The behavior of PINN BALLS is shown in the rightmost plot of Figure 3. Once more, increasing the number of PINN Balls increases the accuracy of the model. This result is non-trivial for a DD-based approach, due to the strong nonlinearity of Navier Stokes' equations and the presence of the pressure, which is a nonlocal quantity. This favorable behavior is due to the global nature of the LM update, which has the scaling advantages of DD, while keeping the invaluable benefits of a second-order method, which is crucial for solving nonlinear PDEs with PINNs [4]. Notably, in our GPU implementation, a global LM update results in an out-of-memory error at approximately 2000 parameters, which are not sufficient to capture the time-dependency of the whole

solution. Indeed, capturing phenomena as the vortex shedding in the cylinder wake requires a model with a consistent number of parameters.

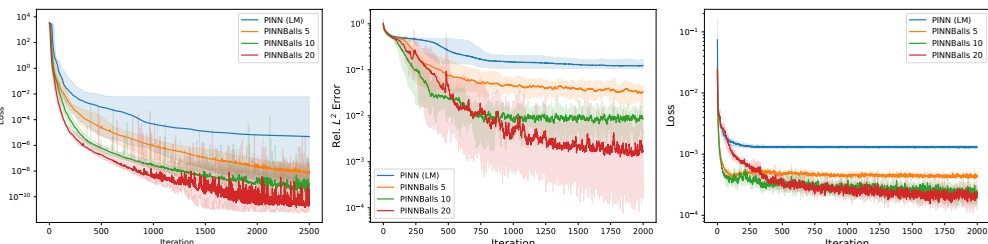

Figure 3: Median Loss value and minimax range over 10 independent trainings of PINN BALLS with an increasing number of submodels on Helmholtz (left), Burgers (center) and Navier Stokes' (right) equations. Each submodel has 3 hidden layers and 10 neurons per layer.

### 5.1.1 Behavior of the Domain Decomposition

Hereinafter, we highlight the behavior of the DD under the chosen initialization and optimization pipeline. Under reasonable conditions, model performance is not highly sensitive to the initial number or placement of basis functions. However, careful initialization is essential for preserving the sparsity structure of the Jacobian $D_\theta L$. Excessively many or poorly placed basis functions can result in significant computational overhead when constructing the matrix $D_\theta L^T D_\theta L$. Utilizing a first-order optimizer for the global update generally yields slow convergence for the majority of the PDE considered. On the other hand, in case of Navier-Stokes equation, convergence is not achieved as the predicted pressure oscillates during training and oftentimes diverges.

Furthermore, our initialization procedure yields a probability density function $p_\phi$ that is approximately uniform. In relatively simple or linear scenarios, such as the Helmholtz equation, we observe that the domain decomposition remains mostly unchanged during training, which aligns with theoretical convergence expectations. However, in nonlinear settings, such as the Navier-Stokes equations, the AAS paradigm introduces meaningful deviations from the uniform distribution. These deviations serve to allocate more submodels in regions with higher solution complexity (e.g., steep gradients) and can be observed in the results in Appendix B. While theoretical convergence to a uniform distribution assumes that the residuals vanish entirely — a reasonable but idealized assumption — this is rarely the case in practice. Consequently, the final distribution does not always converge to uniformity, but this deviation often enhances the model's accuracy. We view this discrepancy between theory and practice as an important and interesting topic that merits deeper exploration in future work.

### 5.2 Comparison with existing approaches

We compare the performance of PINN BALLS with the XPINN and the APINN through the results reported in the respective publications. We train PINN BALLS of variable ensemble size that maintain a comparable number of parameters – roughly 5000 each model –. Comparing with existing architectures, the results in Table 2 clearly highlight the superiority of our model by a large margin in the Helmholtz equation. Notably, this is achieved with a lower number of parameters with respect to the two models highlighted. Regarding Burgers equation, we largely outperform the FBPINN model as they showcase an error on the order of $10^{-1}$. The results obtained by PINN BALLS is comparable to that of the APINN architecture. However, it is important to mention that the models used for comparison directly instantiate a DD that divides the domain in $x = 0$, which is where the shock locates in the solution of Burgers' equation, and represents a somewhat optimal division of the domain – as it captures a discontinuity while maintaining continuous submodel predictions–. Our model, on the other hand, learns the decomposition autonomously. Opposed to the results observed in Figure 3, larger ensembles do not necessarily outperform smaller ones for a fixed parameter count. In practice, it is important that each submodel mantains a reasonable number of parameters to capture the behavior of the target PDE solution in the related subdomain.

Table 2: Comparison of the relative $L^2$ Error achieved by PINN BALLS, XPINN and APINN models on Helmholtz and Burgers Equation, for a comparable number of total parameters.

| Model | Rel. $L^2$ (Burger) | Rel. $L^2$ (Helm.) |
|---|---|---|
| XPINN | $1.3e{-}1 \pm 7.2e{-}3$ | $1.2e{-}3 \pm 1.7e{-}4$ |
| APINN | $9.1e{-}4 \pm 3.6e{-}4$ | $1.2e{-}3 \pm 4.7e{-}4$ |
| PINN BALLS (5) | $\mathbf{2.3e{-}4 \pm 3.3e{-}4}$ | $2.8e{-}9 \pm 1.8e{-}9$ |
| PINN BALLS (20) | $1.7e{-}3 \pm 2.0e{-}3$ | $\mathbf{1.3e{-}9 \pm 1.4e{-}9}$ |

## 5.3 Limitations

While PINN BALLS offer a memory-efficient alternative for second-order training of PINNs, they also present some limitations. Efficiently computing of the Jacobian matrix remains challenging. While sparsity accelerates matrix operations at larger scales, the overhead of assemblying and managing the sparse Jacobian becomes non-negligible at moderate problem size. Therefore, it is crucial to restrict the application of PINN BALLS to cases where the usage of a large number of model parameters is a strict requirement. Furthermore, to the best of our knowledge, existing automatic-differentiation packages do not support backpropagation through sparse matrices. Indeed, the applicability to large industrial use cases assumes the availability of efficient sparse solvers, which although mature, are still generally less optimized for GPU than dense linear algebra routines.

Another limitation lies in the optimization dynamics. Although the sparse structure of the Jacobian improves memory efficiency, it can lead to slow convergence if the individual experts do not have enough parameters or are poorly initialized. In our experiment, we noticed that the training dynamics of the DD strongly depends on the initialization. While this may be attributed to the imbalanced training – first order for the DD and second-order for the PINN BALLS –, properly adapting the DD to highly complex domains can be crucial in various applications since each PINN Ball is responsible for a narrow portion of the input space. We noticed that the quality of the LM optimizer largely bypasses these issues in the DD training. However, further analysis is required to match the quality of the training step of the DD to the effectiveness of the LM step for the PINN BALLS' submodels.

## 6 Conclusion

This paper introduced PINN BALLS, an efficient, scalable, second-order MoE for solving challenging PDEs in general domains. We achieve favorable scaling properties in the number of parameters using DD to construct an ensemble of local models. These models are effectively trained using second-order methods, while the DD partition is updated employing AAS, a framework for which we establish novel existence and stability theoretical results. PINN BALLS achieves state-of-the-art performance on challenging PINN benchmarks, showing an improved model accuracy as the number of submodels increases, and effectively addressing the memory requirements of the second-order methods without compromising their efficiency. We show that combining several submodels is an effective strategy to achieve high accuracies when facing model size constraints, either due to hardware or memory limitations. Our results hint at a trade-off between the number of submodels and their individual capacity, motivating future research into the dynamical scaling of the number of experts.

Incorporating established PINN enhancements, such as Random Fourier Features [29], Temporal Causality [36], and Curriculum Training [20], could further boost the expressiveness and efficiency of the model, particularly given their strong synergy with second-order training methods [4]. Another critical extension involves exploring anisotropic domain decomposition strategies and alternative energy penalization schemes, both of which are crucial for high-fidelity fluid dynamics simulations [30]. Finally, performing the AAS ascent update with respect to the Wasserstein distance may result in smoother updates [18, 12], improving the convergence of the DD partition scheme in complex PDE scenarios.

**Impact Statement**   The model represented enables second-order training of PINN-styled architectures. Moreover, the theoretical backbone ensures a uniform coverage of the PDE domain through a fully learnable domain decomposition. This model represents a step towards enabling PINNs as valuable alternative to traditional PDE solution methods, and their inclusion into industrial applications.

## Acknowledgements

We acknowledge fundings from BMW through the ProMotion programme; funding from IKUR_IKA_23/15 project of the Basque Science Foundation, from the Spanish Ministry of Science, Innovation and Universities (Projects PID2023-149195NB-I00 and PID2022-137442NB-I00), and the Basque Government (Grant Nos. KK-2023/00012, KK-2024/00030, and IT1504-22). Partial funding by BERC 2022-2025 program and the Spanish State Research Agency through BCAM Severo Ochoa excellence accreditation CEX2021-0011 42-S/MICIN/AEI/10.13039/501100011033, and through the project PID2024-158994OB-C42 funded by MICIU/AEI/10.13039/501100011033 and cofunded by the European Union is also acknowledged. IM was supported by the Emmy Noether program of the DFG, project number 403056140.

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

# A   Extension of the Adversarial Adaptive Scheme

For simplicity in this Section we will assume without loss of generality that $\Omega \subset \mathbb{R}^{d_{in}}$ is a compact set with volume $|\Omega| = 1$ and a regular boundary. We also define $r(u(x)) := \mathcal{R}u(x) - f(x)$, and will ignore the boundary conditions for simplicity.

## A.1   Gradient penalization

We consider the problem of this paper in an abstract form: let $U$ be a space of functions $\Omega \to \mathbb{R}^{d_{out}}$, and let $V$ be the space of probability densities with weak gradient in $L^2$, i.e. $V = \mathcal{P}(\Omega) \cap H^1(\Omega)$, where we identify an absolutely continuous probability measure with its density. Consider the minimax problem

$$\min_{u \in U} \max_{p \in V} \mathcal{J}(u, p), \quad \text{with } \mathcal{J}(u, p) := \int_\Omega r(u(x))^2 p(x) dx - \beta \int_\Omega |\nabla p(x)|^2 dx. \tag{14}$$

*Remark* A.1. Since we want to consider the case where the optimal solution of (4) can be approximated by functions in $U$, it is natural to assume that the infimum of the residuals is zero. Moreover, so that the integration against the $L^2$ function $p$ in (14) is meaningful, we will assume that this infimum is meant in the $L^2$ sense. This assumption is weaker than that in the original paper [33], which assumed $r$ to be a surjection from $U$ to $C_c^\infty(\Omega)$; in the latter case a sequence $\{u_n\}_n$ can be easily constructed whose residuals converge to 0 even uniformly.

**Theorem A.2.** *Assume that* $\inf_{u \in U} \|r^2(u)\|_2 = 0$. *Then the optimal value of the minimax problem* (6) *is 0. In particular, for any sequence* $\{u_n\}_n$ *minimizing* $\|r^2(u)\|_2$ *there exists a corresponding sequence of probability density functions* $\{p_n\}_n$, *each maximizing* $\mathcal{J}(u_n, \cdot)$, *such that*

$$\lim_{n \to \infty} \mathcal{J}(u_n, p_n) = 0. \tag{15}$$

*Moreover, the probability density functions* $\{p_n\}_n$ *converge strongly in* $H^1(\Omega)$ *to the uniform distribution on* $\Omega$.

*Remark* A.3. Theorem A.2 improves upon previous results in at least two aspects. First, it shows convergence to solutions of the minimax problem using the $H^1$ regularization, which is much more widespread and practical than the Lipschitz regularization considered in the theory of [33]. Moreover, we show that the sequence of marginally optimal probability distributions $\{p_n\}_n$ converge to the uniform distribution. This is a general fact, not depending on the specific minimizing sequence $\{u_n\}_n$, and rules out a concentration of $p_n$ on regions with a higher residual as the total error converges to zero.

*Proof of Theorem A.2.* To begin with, note that for a given $u \in U$, a natural lower bound for $\max_p \mathcal{J}(u, p)$ is given by choosing the uniform density $\tilde{p}(x) \equiv 1$, this is:

$$\max_{p \in V} \mathcal{J}(u, p) \geq \mathcal{J}(u, \tilde{p}) = \int_\Omega r^2(u_n(x)) \tilde{p}(x) dx - \beta \int_\Omega |\nabla \tilde{p}(x)|^2 dx = \int_\Omega r^2(u_n(x)) dx. \tag{16}$$

This shows that (14) must be non-negative. Let now $(u_n)_n$ be a minimizing sequence for $\|r^2(u)\|_2$. Fix momentarily a given $n$; we will first show that there exists an optimizer for the maximization problem $\max_{p \in V} \mathcal{J}(u, p)$. Let $p \in V$ be a candidate with a higher or equal score than the uniform density $\tilde{p}$; then, by (16) we must have:

$$\int_\Omega r^2(u_n(x)) dx \leq \int_\Omega r^2(u_n(x)) p(x) dx - \beta \int_\Omega |\nabla p(x)|^2 dx$$

$$\beta \int_\Omega |\nabla p(x)|^2 dx \leq \int_\Omega r^2(u_n(x)) \left(p(x) - 1\right) dx \leq \|p - 1\|_2 \|r^2(u_n)\|_2$$

$$\leq C \|\nabla p\|_2 \|r^2(u_n)\|_2,$$

where $C$ is the constant of Poincaré's inequality [11] (since $p$ is a probability measure, and $\Omega$ has unit volume, the average value of $p$ is 1; also note that $\Omega$ has a regular boundary by assumption). Dividing by the norm of $\nabla p$ then yields:

$$\beta \|\nabla p\|_2 \leq C \|r^2(u_n)\|_2, \tag{17}$$

and using the Poincaré's inequality once again on the left-hand side results in an estimate for the deviation of $p$ with respect to the uniform density:

$$\tfrac{1}{C}\|p-1\|_2 \le \|\nabla p\|_2 \le \tfrac{C}{\beta}\|r^2(u_n)\|_2. \tag{18}$$

We have shown an $H^1$ bound for any probability density that has a higher score than the uniform density $\tilde{p}$ by $\mathcal{J}(u_n,\cdot)$. Consequently, the superlevel sets of $\mathcal{J}(u_n,\cdot)$ are compact for the $H^1(\Omega)$ weak convergence, so given a maximizing sequence $\{p_n^k\}_k$ there exists a (weak) cluster point $p_n$, which must be a maximizer for $\mathcal{J}(u_n,\cdot)$ due to the upper-semicontinuity of $\mathcal{J}(u_n,\cdot)$:

$$\sup_{p\in V}\mathcal{J}(u_n,p) = \limsup_{k\to\infty}\mathcal{J}(u_n,p_n^k) \le \mathcal{J}(u_n,p_n) \le \sup_{p\in V}\mathcal{J}(u_n,p). \tag{19}$$

Finally, since $p_n$ satisfies (18), it holds:

$$\tfrac{1}{C}\|p_n-1\|_2 \le \|\nabla p_n\|_2 \le \tfrac{C}{\beta}\|r^2(u_n)\|_2. \tag{20}$$

Since $r^2(u_n)$ converges to zero in $L^2$, we obtain both strong $L^2$ convergence of $p_n$ to the uniform distribution $\tilde{p} \equiv 1/|\Omega|$ and strong $L^2$ convergence of the gradient $\nabla p_n$ to zero as $n \to \infty$. Combining the convergence behavior of $\{r^2(u_n)\}_n$ and $\{p_n\}_n$ yields (15).

$\square$

## A.2 Entropy penalization

Although intuitive, the gradient penalization term in (14) assumes that the probability density has a gradient, which may exclude some interesting cases. Besides, there is no mechanism preventing $p$ to reach zero, which could imply that the residuals in some regions may not be accounted for at all. Inspired by the field of optimal transport, and in particular Wasserstein gradient flows (see Section A.3 for a discussion), we propose an alternative penalization term that solves these issues. We consider:

$$\min_{u\in U}\max_{p\in\mathcal{P}(\Omega)}\mathcal{J}(u,p), \qquad \text{with } \mathcal{J}(u,p) := \int_\Omega r(u(x))^2 p(x)dx - \beta\int_\Omega p(x)\log p(x)dx. \tag{21}$$

An interesting observation is that in this case the optimizer of the nested maximization step can be given in closed form, given by Lemma A.4. This is an elementary result; we however gather the proof in Section A.4 for completeness.

**Lemma A.4.** *Assume $r(u(x))^2 \in C(\Omega)$. Then the optimizer of the problem*

$$\min_{p\in\mathcal{P}(\Omega)}\beta\int_\Omega p(x)\log p(x)dx - \int_\Omega r(u(x))^2 p(x)dx. \tag{22}$$

*is given by*

$$p^*(x) := C\exp\left(r(u(x))^2/\beta\right), \tag{23}$$

*with $C := (\int_\Omega \exp\left(r(u(x))^2/\beta\right))^{-1}$ a normalization constant ensuring $p^*$ has unit mass.*

Note that an assumption on the regularity of $r^2(u(x))$ is necessary in order to obtain existence of optimizers, though a tighter assumption may in principle be used (e.g. upper lower-semicontinuity and boundedness).

As in the previous Section, we will need to assume certain regularity of the residuals. Since in this case the regularization term only requires $p$ to be a probability distribution (which is a rather weak assumption), we need a stronger setting for $r^2(u)$. In this case, the appropriate assumption is that $\inf_u \|r^2(u)\|_{C(\Omega)} = 0$, this is, along at least some sequence, the residuals are continuous and converge uniformly to zero. Again, this assumption is weaker than the assumptions in [33], since the latter implies the existence of some $u^*$ such that $r^2(u^*) \equiv 0$.

**Theorem A.5.** *Assume that $\inf_{u \in U} \|r^2(u)\|_{C(\Omega)} = 0$. Then the optimal value of the minimax problem* (21) *is 0. In particular, for any sequence $\{u_n\}_n$ minimizing $\|r^2(u)\|_{C(\Omega)}$ there exists a corresponding sequence of probability density functions $\{p_n\}_n$, each maximizing $\mathcal{J}(u_n, \cdot)$, such that*

$$\lim_{n \to \infty} \mathcal{J}(u_n, p_n) = 0. \tag{24}$$

*Moreover, the probability density functions $\{p_n\}_n$ are everywhere positive and converge* uniformly *to the uniform distribution on $\Omega$.*

*Proof.* First note that, as in the proof of Theorem A.2, a natural lower bound for $\max_p \mathcal{J}(u, p)$ is given by choosing the uniform density $\tilde{p}(x) \equiv 1$. Therefore the optimum of (21) must be non-negative.

Now consider $\{u_n\}_n$ a minimizing sequence for $\|r^2(u)\|_{C(\Omega)}$. Under the assumptions of the theorem, by Lemma A.4 each $u_n$ is associated with the optimal probability distribution given by (23), that we denote by $p_n$ (with a corresponding normalization constant $C_n = 1/\int_\Omega \exp(r^2(u_n(x))/\beta)$). Then the minimax problem becomes:

$$\max_{p \in \mathcal{P}(\Omega)} \mathcal{J}(u_n, p) = \int_\Omega p_n(x)[r^2(u_n(x)) - \beta \log p_n(x)]dx$$

$$= \int_\Omega p_n(x)[r^2(u_n(x)) - \beta \log C_n - r^2(u_n(x))]dx = -\beta \log C_n, \tag{25}$$

where we used that $p_n$ has unit mass. Finally let us bound (25): noting that $1 \le \exp(r^2(u_n(x))/\beta) \le \exp(\|r^2(u_n)\|_\infty/\beta)$,

$$1 = \int_\Omega 1 dx \le \overbrace{\int_\Omega \exp(r^2(u_n(x))/\beta)\,dx}^{1/C_n} \le \int_\Omega \exp(\|r^2(u_n)\|_\infty/\beta)dx = \exp(\|r^2(u_n)\|_\infty/\beta),$$

which after taking the logarithm becomes:

$$0 \le -\log C_n \le \frac{\|r^2(u_n)\|_\infty}{\beta}. \tag{26}$$

Plugging this bound into (25) and using that $\|r^2(u_n)\|_\infty \to 0$ concludes that the minimax optimal value (21) is zero. The associated $C_n$ converges to 1, since $r^2(u_n)$ converges uniformly to zero, and therefore $p_n$ converges likewise uniformly to the uniform density. $\qquad \square$

### A.3 Relation to Wasserstein gradient flows

The theory of gradient flows deals with the generalization of curves of steepest descent of the form

$$\dot{x} = -\nabla F(x) \tag{27}$$

to general metric spaces, where neither the left-hand side nor the right-hand side may be well defined. This framework has proven to be extremely fruitful in the field of PDEs, since the founding work of Jordan, Kinderlehrer and Otto [18, 26] showed that many PDEs of interest can be described as curves of steepest descent with respect to certain functionals the space of probability measures endowed with the Wasserstein distance (see e.g. [34, 1] for an overview).

To allow for a general formulation in this abstract framework, it is customary to employ the formalism of *minimizing movements*. This is, for a metric space $(M, d)$, a functional $F : M \mapsto \mathbb{R} \cup \{+\infty\}$, an initial condition $x_0 \in M$ and a time step $\tau$ one considers the discrete trajectory generated by:

$$x_\tau^0 = x^0, \qquad x_\tau^k \in \arg\min_{x \in M} \frac{1}{2\tau} d(x, x_\tau^{k-1})^2 + F(x), \quad k = 1, 2, \ldots \tag{28}$$

When $(M, d)$ is a finite-dimensional Euclidean space, (28) can be shown to reduce to an implicit Euler scheme approximating (27). But the benefit of this general framework stems from its application to functional or measure spaces, as well as its ability to handle non-smooth functionals. Then, under suitable assumptions on $F$, $M$ and $d$, the discrete trajectories can be shown to converge to a limit trajectory, which can then be characterized with tools from the theory of partial differential equations.

In relation to the matter of this paper, we have considered two regularization terms: the Dirichlet energy $G_D(p) = \int_\Omega |\nabla p|^2 dx$ and the entropy $G_E(p) = \int_\Omega p(x) \log p(x) dx$. The gradient flow of the Dirichlet energy with respect to the $L^2(\Omega)$ topology yields the heat equation [18]. The heat equation tends to homogenize a given initial condition, which is a desired property in the minimax scheme we studied. Interestingly, the heat equation can be also obtained as the gradient flow of the entropy $G_E$, in this case on the space of probability measures $\mathcal{P}(\Omega)$ endowed with the $L^2$-Wasserstein distance [18]. Subsequently, we can expect both functionals to perform similarly in a gradient-descent framework, as long as gradients are computed with respect to the right metric (this is, $L^2$ for $G_D$ and $L^2$-Wasserstein for $G_E$). It remains somewhat surprising that in our numerical experiments the entropy seems to be better behaved even using the importance sampling update rule (13), which has an $L^2$ flavour. We consider this further proof of the robustness of the method, and plan to investigate purely Wasserstein updates for the probability mesaure $p$ in future work.

### A.4 Additional proofs

*Proof of Lemma A.4.* For simplicity let us define $\mathcal{E}(p) := \beta \int_\Omega p(x) \log p(x) - r^2(u(x))p(x)$. A minimizer $p^*$ exists by sheer lower-semicontinuity of $\mathcal{E}$ and compactness of $\mathcal{P}(\Omega)$ with respect to the weak* topology. Moreover, any minimizer must have a positive density; otherwise the objective value would be infinite, since the uniform density is a feasible candidate with a finite objective. This means that by slight abuse of notation we can use $p^*$ also to refer to its density.

Let $\theta$ be an arbitrary function in $L^\infty(\Omega, p^*)$ satistying $\int_\Omega \theta p^* = 0$, and consider the perturbation of $p^*$ given by $p_\varepsilon := (1 + \varepsilon\theta)p^*$. It follows that for small $\varepsilon$, $p_\varepsilon$ remains in $\mathcal{P}(\Omega)$. Let us compute how $\mathcal{E}(p^*)$ changes under a perturbation:

$$0 \le \frac{\mathcal{E}(p_\varepsilon) - \mathcal{E}(p^*)}{\varepsilon} = \beta \int_\Omega \frac{p_\varepsilon(x) \log p_\varepsilon(x) - p^*(x) \log p^*(x)}{\varepsilon} dx - \int_\Omega r^2(u(x))\theta(x)p^*(x)dx,$$

which, using the monotonicity of the function $s \mapsto s \log(s)$, by dominated convergence yields in the limit $\varepsilon \to 0$:

$$0 \le \int_\Omega \theta(x)p^*(x)[\beta \log p^*(x) - r^2(u(x))]. \tag{29}$$

Since (29) must hold also by replacing $\theta$ by $-\theta$, the inequality turns into an equality. Moreover, since $\theta p^*$ is an arbitrary zero-mean $L^\infty$ function, the expression between brackets must be constant. All in all, parametrizing this constant offset by $\beta \log C$, with $C > 0$, yields:

$$\beta \log p^*(x) - r^2(u(x)) = \beta \log C \Rightarrow p^* = C \exp(r^2(u(x))/\beta). \tag{30}$$

Thus we identify $C$ as a normalization constant enforcing that $p^*$ is a probability distribution, i.e.

$$C = \left( \int_\Omega \exp(r^2(u(x))/\beta) \right)^{-1}.$$

The boundedness of $r^2(u)$ guarantees that $C$ is finite, and that $p^*$ is bounded away from zero, while focusing the attention on the regions with a larger residual. Whenever the residual is uniform, $p^*$ will become uniform as well. □

## B  Details of the PDEs

### B.1  Helmholtz Equation

The Helmholtz equation is a diffusive-reactive second-order equation similar to the wave equation. Although linear, the Helmholtz equation represents a challenging PDE instance for PINNs due to the high-frequency components included in its solution. In our paper, we choose to solve the bidimensional formulation, which is highlighted in Equation (31) in the domain $\Omega = [0, 1]^2$

$$\begin{aligned} \Delta u + k_x k_y u &= f, & x, y \in \Omega \\ u(x, y) &= 0, & x, y \in \partial\Omega \end{aligned} \tag{31}$$

with $f(x, y) = -k_x k_y \sin(k_x x) \sin(k_y y)$. For our test case we choose $k_x = 4$ and $k_y = 1$, analogous to that presented in [14] for comparability. Each submodel consists of as little as 2 hidden

layers with 10 neurons each. Training is performed with $N_r = 10^4$ collocation points for training the PDE residuals, initially sampled with latin hypercube sampling, and $N_b = 3 \cdot 10^3$ points for training the boundary and initial condition in $\partial \Omega$. We train PINN BALLS with 2 hidden layers and 10 neurons per layer for 2000 iterations, each iterations consists on a single LM step on $\theta$ and 20 Adam iterations on $p_\phi$. Figure 4 showcases the result of the best performing model on Helmholtz equation, alongside the learnt DD.

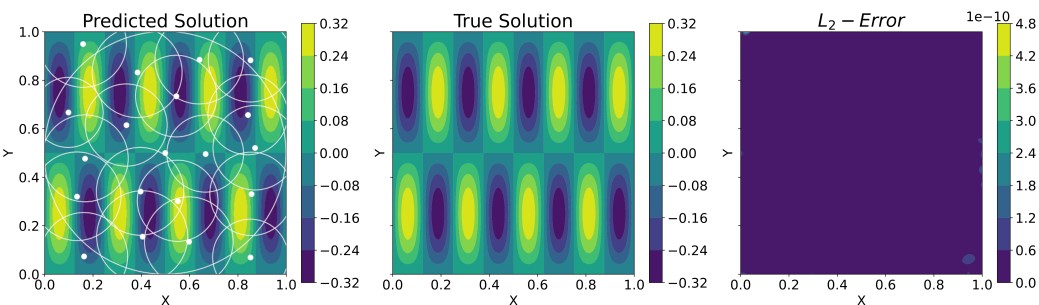

Figure 4: Prediction with learnt center and radii of the PINN Balls (left); correct solution (center); and $L^2$ Error achieved by our best performing model on Helmholtz equation.

## B.2 Burgers' Equation

Burgers' equation is a 1D version of Navier-Stokes equations without pressure. Its solution at high times present a discontinuity, which makes it challenging for spectrally biased architectures. The specific instance chosen in our numerics for Burgers' equation is the same as in [27]. In particular, we refer to the exact same data provided by the authors. In particular, given $(x, \tau) \in \Omega = [-1, 1] \times [0, 1]$, we solve for $u : \Omega \to \mathbb{R}$ the following equation:

$$
\begin{aligned}
\partial_\tau u + u \partial_x u - \nu \partial_x^2 u &= 0, & (x, \tau) \in \Omega, \\
u(x, 0) &= -\sin(\pi x), & x \in [-1, 1], \\
u(-1, \tau) = u(1, \tau) &= 0, & \tau \in [0, 1],
\end{aligned}
\tag{32}
$$

with the diffusivity $\nu$ being equal to $\frac{0.01}{\pi}$ for this specific instance. The correct solution is provided publicly by the authors of [27]. Training is performed with $N_r = 10^4$ collocation points for training the PDE residuals, initially sampled with latin hypercube sampling, and $N_b = 3 \cdot 10^3$ points for training the boundary and initial condition in $\partial \Omega$. We train PINN BALLS with 3 hidden layers and 10 neurons per layer for 2000 iterations, each iterations consists on a single LM step on $\theta$ and 20 Adam iterations on $p_\phi$. Figure 5 showcases the result of the best performing model on Burgers' equation, alongside the learnt DD.

## B.3 Navier-Stokes Equation

The last and most crucial PDE instance considered is given by the incompressible Navier-Stokes equations. In particular, solve the fluid flow past a 2D cilinder presented in [17]. We consider the same instance proposed in the aforementioned paper, given by $(x, y, t) \in \Omega = [-2.5, 7.5] \times [-2.5, 2.5] \times [0, 16]$ with the goal of finding $\vec{u} : \Omega \to \mathbb{R}^3$, defined as $\vec{u}(x, y, t) = [u(x, y, t), v(x, y, t), p(x, y, t)]^T$. In this formulation $u$ and $v$ represents respectively the horizontal and vertical components of the fluid

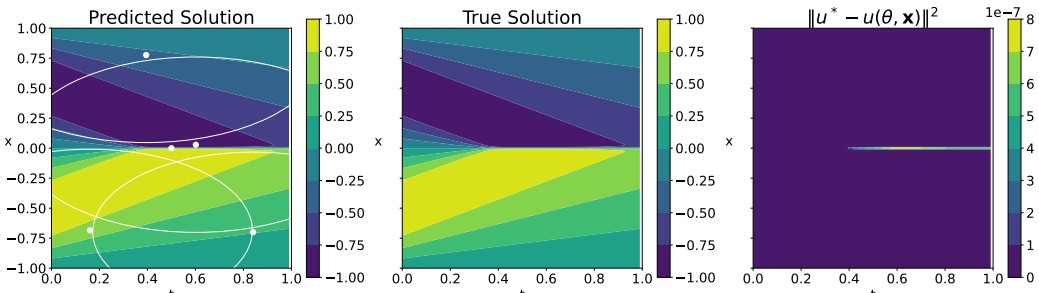

Figure 5: Prediction with learnt center and radii of the PINN Balls (left); correct solution (center); and $L^2$ Error achieved by our best performing model on Burgers' equation.

velocity; and $p$ the pressure. Navier-Stokes equations are expressed in vectorized form as follows:

$$
\begin{aligned}
&\partial_t u + u\partial_x u + v\partial_y u - \frac{1}{Re}\left(\partial_x^2 u + \partial_y^2 u\right) + \partial_x p = 0, \quad (x,y,t) \in \Omega, \\
&\partial_t v + u\partial_x v + v\partial_y v - \frac{1}{Re}\left(\partial_x^2 v + \partial_y^2 v\right) + \partial_y p = 0, \quad (x,y,t) \in \Omega, \\
&\partial_x u + \partial_y v = 0, \qquad\quad (x,y,t) \in \Omega, \\
&u(x,y,0) = g_{u_0}(x,y), \quad (x,y) \in [-2.5, 7.5] \times [-2.5, 2.5], \\
&v(x,y,0) = g_{v_0}(x,y), \quad (x,y) \in [-2.5, 7.5] \times [-2.5, 2.5], \\
&u(2.5, y, t) = 1, \qquad\quad (y,t) \in [-2.5, 2.5] \times [0, 16], \\
&v(2.5, y, t) = 0, \qquad\quad (y,t) \in [-2.5, 2.5] \times [0, 16],
\end{aligned}
\tag{33}
$$

where $Re$ is the Reynolds' number, which is an adimensional quantity defined by the problem and is set to 100 for our case. The initial conditions $(g_{u_0}, g_{v_0})$ can be found in the repository of [27], as well as an high fidelity solution which is used as ground truth. At $x = -2.5$ the fluid velocity at the inlet is imposed. Further conditions are given by the presence of a cylinder centered in $(x, y) = (0, 0)$ with radius 0.25. Furthermore, an additional condition appears at the borders, namely where $y = \pm 2.5$, where the no-slip condition can be chosen ($u = v = 0$) or the correct solution can be given as boundary condition. Since the simulation provided in [27] refers to a free-flow stream, we use the correct solution at the boundaries.

To train our PINNs, we use $N_r = 5 \cdot 10^5$ collocation points for training the PDE residuals, sampled with latin hypercube sampling, and $N_b = 2 \cdot 10^4$ points for training the boundary and initial condition in $\partial\Omega$. Morever, at every iteration, we minimize the loss on random batches of the training data, respectively $10^4$ points for the residuals and $5 \cdot 10^3$ for boundary and initial condition. Figure 6 showcases the result of the best performing model on Navier-Stokes' equations, alongside the learnt DD. Notice that we do not normalize the pressure predicted by the model, but we do compute the absolute error by comparing the normalized pressures.

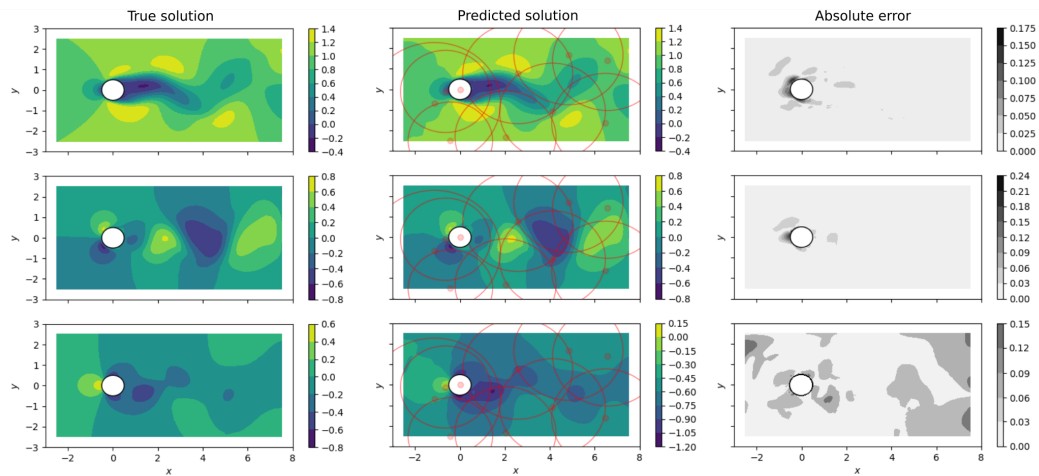

Figure 6: Prediction with learnt center and radii of the PINN Balls (left); correct solution (center); and Absolute Error achieved by our best performing model on Navier Stokes' equations. The top row refers to the horizontal velocity $u$, the central row to the vertical velocity $v$, and the bottom row to the pressure $p$.

