# OpenReview forum: "PINN Balls: Scaling Second-Order Methods for PINNs with Domain Decomposition and Adaptive Sampling"
_NeurIPS.cc/2025/Conference — NeurIPS 2025 poster_

### Official Review · Reviewer_MZup · 2025-07-01

**Clarity:** 3
**Significance:** 3
**Originality:** 3
**Rating:** 4
**Confidence:** 4

**Summary:**

The submission suggests algorithmic improvements to physics-informed neural networks (PINNs): it proposes to combine domain decomposition, adaptive sampling, and second-order optimisation.
Specifically, the approach is to replace a single neural network with an ensemble, combining the outputs of each ensemble member using a locally-supported weighting function (akin to a partition of unity).
The collocation points for the PINN are sampled from the same weighing function, and the resulting model is trained using the Levenberg--Marquardt optimiser.
The submission also includes an analysis of the existence of solutions to the minimax problem underlying the adaptive sampling algorithm, which extends a previous result. Numerical results show that the resulting algorithm is competitive with prior approaches.

**Questions:**

Here is a collection of some more minor questions. They don't affect my score, but they might be helpful feedback for revising the paper. (I'd still be interested in their answers, though I understand that rebuttal time and space may be limited.)

1. Line 187: "Ensuring \phi > 0 can be done by choosing a sufficiently large $s$ at initialisation" -> is this always true?
2. Equation 10: Does this loss assume the negative entropy regularisation?
3. Equation 10 (again): How does the domain decomposition interact with the boundary?
4. Equation 11: I thought Equation 10 ("J") was the preferred loss, not Equation 4 ("L"). What have I missed? Same in Line 228.
5. Line 241: The exponential decay in learning rate feels quite aggressive. How important is that? And what kind of exponential decay is used?
6. Line 245: What's the domain of the problem? The target function $f(x) = sin(4x)sin(y)$ has 4 "bumps" if x and y are between 0 and $\pi$. Depending on the domain, should there be a "correct ensemble size" of four, or is the number of ensembles less straightforward to imagine?
7. Line 247: What are $K$-means applied to?
8. Line 266: Are those results in the paper? I can't find them, but I would like to see them.
9. Line 268: To what extent is the condition number of $DL$ relevant for Figure 1? Doesn't the experiment measure a single step of Levenberg-Marquardt implemented with a direct solver? In my understanding, this should be independent of any condition numbers. Have I missed something?

**Ethical Concerns:**

["NO or VERY MINOR ethics concerns only"]

**Final Justification:**

My original assessement gave a score of 4, mainly listing experimental clarity and gaps in the numerical-linear-algebra discussion as weaknesses. The rebuttal alleviated some of these concerns, but a significant part of the numerical-linear-algebra-related criticism persists, which is why the evaluation remains a 4. Details are in the reply to the rebuttal.

**Limitations:**

yes

**Paper Formatting Concerns:**

-

**Quality:**

3

**Strengths And Weaknesses:**

## Strengths
The submission is a nice paper, and I enjoyed reading it. It does not just contain a well-explained combination of ideas; merging domain decomposition and adaptive sampling appears to be effective, and the resulting algorithm is sufficiently simple to motivate reimplementation in future PINN studies. Furthermore, the theory is novel, convincing, and a valuable addition to the literature on adaptive sampling for PINNs.

## Weaknesses
While I recommend accepting the paper, I am currently assigning a borderline score due to the weaknesses explained below. I am open to revising my evaluation if my concerns turn out to be easy to resolve.
### Second-order optimisation

The second-order optimisation parts of the paper are not as convincing as the other components. The main optimisation contribution of the paper is replacing [33]'s stochastic gradient descent with Levenberg--Marquardt. However, the paper does not demonstrate that for the PINN-Balls, second-order methods outperform first-order methods. Instead, the motivation for choosing second-order methods is that they are promising for PINNs, as indicated by recent studies [4, 24].
- If using second-order optimisers for PINN-Balls is considered a contribution of the paper, there needs to be a corresponding benchmark or analysis.
- If referencing [4, 24] is considered a sufficient argument for using second-order methods, then I think that using second-order methods for PINN-balls is not a contribution of this paper.

Section 4.1 is dedicated to discussing the sparsity of the Gauss--Newton matrix $D L^\top DL$ with the corresponding savings in memory and runtime. This perspective overlooks matrix-free linear solvers, such as conjugate gradients, which do not need sparsity as long as matrix-vector products are efficient. For $G=  DL^\top DL$, these matrix-vector products can be implemented with automatic differentiation. I believe this omission is a weakness in Section 4.1; a discussion or benchmark of alternative linear algebra approaches would be appropriate.

### Experimental clarity

The numerical experiments leave several open questions, and I believe this hinders the appreciation of the results to the extent they should be appreciated. For example, I would like the following information added to the context of Figures 1 and 2 and Table 2:
- Figure 1:
    - The sparsity of the $DL^\top DL$ matrix should depend on the distribution of points, so it should vary quite a lot during optimisation. Why is this experiment, which benchmarks only a single Levenberg--Marquardt step, representative of the sparsity of the system matrix?
    - All but the left-centre plot would be much easier to read on a logarithmic scale. As is, I find trends difficult to identify.
    - The far-left figure suggests that the sparsity does not yield any runtime improvements. Is this because the problem is too small? What happens to larger matrices?
    - All those numbers are on CPU, do I understand correctly? Do the authors have any intuition about what they look like on GPU? In my experience, CPU and GPU runtimes differ considerably when running linear algebra routines.
    - Do I read the figure correctly that the more PINN balls, the more efficient the linear algebra? How many balls are "too many" in terms of approximation quality?
- Figure 2:
    - What does $L^2$ error mean here? Is this a comparison to a reference solution or the $L^2$ norm of the residual? If it is the former, where does the reference come from?
    - What precision is used? L2 errors of $10^{-10}$ sound like double precision, but in my experience, this is difficult on GPUs.
    - It appears that the more PINN balls are used, the lower the loss. What happens to more than 20 Pinn balls?
- Table 2: The results seem to disagree with Figure 2 in the sense that in Figure 2, 20 balls achieve lower error than five balls, but in Table 2, the roles are reversed. Where does this discrepancy come from?

I list some additional, more minor questions under "Questions" below. (For reference, I don't expect all to be answered in the rebuttal. It would be nice, but I understand that time and space for the rebuttal are limited.)

Despite these issues, the paper's strengths outweigh the weaknesses, and I recommend acceptance. However, resolving the issues I discuss above (or explaining how I misunderstood something) would lead me to improve my score.

---

> ### Author Rebuttal · Authors · 2025-07-30
>
> We sincerely thank the reviewer for the positive feedback, appreciation of our work, and the thorough and constructive comments, which we believe will significantly enhance the quality of the manuscript. First, we address the reviewer’s major concerns in detail.
>
> **Second Order Optimization:**
> * As the reviewer rightly noted, the primary contribution of our work is not the use of second-order optimization per se, but rather the development of a framework that enables such optimization to scale beyond current memory limitations. While prior work has shown that second-order methods enables accurate training of PINNs even in low parameter regimes, we are the first to propose a strategy which can extend this framework to a consistent number of parameters, which are at times necessary to predict certain PDE solutions. For instance, in the case of the time dependent Navier-Stokes equations, the computation of the Jacobian $D_\theta L$ leads to an out-of-memory error at approximately 2000 parameters on our GPU implementation. This is due to the need to backpropagate through each collocation point and each PDE equation. However, in our experience, 2000 parameters are insufficient to model complex behaviors such as vortex shedding behind a cylinder. The combination of a compactly supported RBF-based domain decomposition — which induces sparsity in the Jacobian — and a sparse solver for the LM step — which keeps the optimization computationally feasible — forms the core of our contribution. Together, these components enable the use of second-order training for physics-informed models in settings where high model capacity is not just beneficial, but necessary. An additional reason why we reference [4, 24] is to highlight the general dominance in performance of second-order optimization with respect to first order optimizers. Nevertheless, we have tested first-order optimization and found that it is consistently outperformed by second-order approaches in all cases, as in the cited papers. For the Navier-Stokes equations in particular, first-order optimization fails to converge and exhibits significant oscillatory behavior - especially in the pressure field - even after 500'000 iterations. These findings will be emphasized more clearly in the revised version of the manuscript.
>
> We appreciate the reviewer’s suggestion regarding matrix-free solvers such as conjugate gradient. In principle, these methods are attractive due to their memory efficiency. However, the construction of the Jacobian $D_\theta L$ can trigger itself out-of-memory errors with as few as a few thousand parameters. However, matrix-free methods could benefit from the sparse structure introduced by our method, as it enables efficient sparse matrix-vector products that are central to such solvers. We will extend Section 4.1 to incorporate this valuable point raised by the reviewer and more clearly position our method in relation to matrix-free approaches.
>
> **Experimental Clarity**
> * **Figure 1**
> * As noted by the reviewer, the domain decomposition may alter the sparsity structure of the Jacobian over the course of training due to changes in the basis radii. If preserving high sparsity is critical, the basis radius can be capped or even fixed after initialization. The figure in its current form provides insight into the sparsity pattern at initialization, which can be an indicator of the sparsity during training, when required.
> * We chose to use a linear scale in Figure 1 to better highlight trends in the high-parameter regime, which is the primary focus of the paper. While a logarithmic scale offers finer resolution for lower parameter counts, we believe the linear scale more clearly conveys the relevant trends for our purposes.
> * It is true that sparse solvers are generally slower than their dense counterparts, particularly due to limitations in available open-source implementations. However, as also noted in the Limitations section, the advantage of sparse solvers lies in their scalability and memory efficiency. Dense solvers often run out of memory before solving large systems—whereas the sparse solver we used successfully handled systems with over 10,000 parameters. We will clarify in the revised manuscript that the dense solver in our experimental setup fails before 8000 parameters, whereas the sparse solver remains tractable.
> * The experiments were conducted on CPU to ensure fair comparison, given the lack of GPU-native, open-source sparse solvers. While GPU acceleration would benefit both dense and sparse solvers, the performance advantage may differ based on implementation efficiency and the size of the problems considered (typically GPU advantage will be more noteworthy for larger problems, both for dense and sparse solvers).
> * Indeed, sparse solvers perform better when the matrices involved are more sparse. However, using too many submodels (PINN Balls) for a fixed total number of parameters reduces the expressivity of each submodel. Quantitatively, we observed a drop in performance when submodels are only able to capture low complexity (e.g., one or two layers with at most eight neurons per layer).
>
> * **Figure 2**
> * Two of the subfigures report the Loss value (so the norm of the residuals) since the $L^2$ error against the exact solution is computationally expensive to evaluate at every iteration — especially for the Navier-Stokes equation. However, we provide the error against the exact solution for Burgers' equation, since the norm of the residuals is at times misleading in that case. Reference solutions for Burgers’ and Navier-Stokes equations are computed trhough high-fidelty simulations and are available in several public GitHub repositories, referenced in the Appendix.
> * The optimization is indeed performed using double precision. This choice is particularly important for second-order methods, where precision is crucial. This aspect further enhances the challenge in using a large number of parameters.
> * Increasing the number of PINN Balls would further increase the accuracy of the model. Considering a limit scenario, where an infinite number of PINN Balls is used, the number of parameters of the model would be infinite too. In this scenario, the universal approximation theorem holds, and it should be possible to fit a PDE solution with arbitrary accuracy. However, this is always subject to numerical limitations. Regarding the specific PDEs included in our paper, for the case of Helmholtz equation, we do not expect the loss to decrease for more than 20 PINN Balls, since the error is already approaching machine precision. However, for the cases of Navier Stokes' and Burgers' equations, it is likely that using a larger number of PINN Balls will further increase the accuracy of the overall model.  Finally, note that increasing the number of PINN Balls entails an increase in the numerical complexity of the training step, so one needs to consider the trade-off between accuracy and computational cost, which in our case is reasonably satisfactory at 20 PINN Balls.
>
> **Table 2** The observed discrepancy between PINN Balls 5 and PINN Balls 20 is primarily due to architectural choices. To ensure fair comparison with XPINN and APINN, we constrained both models to have approximately the same number of parameters. In this configuration, the submodels of PINN Balls 20 appeared to be too small for this scenario - roughly as those mentioned in the answer above -.
>
> We hope that our responses clarified the major doubts of the reviewer. We now address the minor concerns in a more compact way:
> 1) Yes. In practice, when using K-means clustering, the exact value of sigma that ensures $\phi >0$ for every collocation point is given by the standard deviation of each cluster.
> 2) Yes, the loss function in Equation 10 is based on the negative entropy.
> 3) The loss term for the boundary condition is evaluated on points sampled uniformly over the boundary, and as such it does not interact directly with the domain decomposition in the minimax problem. The mechanism by which boundary errors affect the DD is that high errors at the boundary propagate to the domain, which trigger changes in the DD.  As we show this results in an efficient training; alternatively, one can also impose the output of PINN Balls to match the exact boundary conditions, as done in several existing works on PINNs.
> 4) The use of $L$ instead of $J$ is indeed a typo. We will correct this in the revised version.
> 5) The exponential decay is important for maximizing the performance of the LM optimization in late training stages. The core concept behind the exponential decay is to conclude the optimization with few LM iterations free from the AAS maximization step. We use the exponential decay implemented in optax, with end value $10^{-10}$.
> 6) The problem is indeed solved over the domain $[0, \pi]$. We agree with the reviewer that an optimal ensemble size likely exists, although it is difficult to predict a priori for general PDEs. Identifying the optimal number of submodels — perhaps via progressively adding submodels during training — is an important topic for future investigation.
> 7) K-means is applied to the collocation points in the PDE domain.
> 8) We did not include those results as they appeared nearly identical to the current ones.
> 9) In our intuition, a sparse ill-conditioned matrix could yield a dense inverse matrix, which would hinder the effectiveness of the sparse solver. Notably, the Jacobian of a PINN is often ill-conditioned, especially for nonlinear PDEs. However, we have not noticed this issue numerically, attributing this to the regularization of the LM step. However, this could also suggest that the condition number may not play a decisive role in this context, as per the reviewer's intuition.
>
> We thank the reviewer once again for the thoughtful feedback and constructive suggestions. Their input has contributed meaningfully to improving the clarity, rigor, and completeness of our work.

---

> > ### Comment · Reviewer_MZup · 2025-08-01
> >
> > Thank you for the helpful response! I also appreciate that each of my points has been replied to.
> >
> > However, I believe my original criticism pertaining to the relation to matrix-free linear algebra still holds. On the one hand, I agree that dense linear algebra wouldn't scale because Jacobians can be large. On the other hand, the rebuttal writes:
> >
> > > However, matrix-free methods could benefit from the sparse structure introduced by our method, as it enables efficient sparse matrix-vector products that are central to such solvers. We will extend Section 4.1 to incorporate this valuable point raised by the reviewer and more clearly position our method in relation to matrix-free approaches.
> >
> > I agree with the perspective on matrix-free linear algebra like CG and I appreciate that a discussion is added to the paper. However, additional positioning in Section 4.1 is not enough change for me to increase my score. To achieve this, I would have expected an explanation that shows how something like CG can't be as effective as the sparse solver, or a benchmark between sparse solvers and matrix-free solvers in the PINN-ball context. As is, I believe my concerns on the numerical-linear-algebra component persist.
> >
> > Thank you also for the explanations about the experimental clarity. It would have been nice to have more clearly stated how the submission will be changed, but I will assume that all clarifications will end up in the paper somehow. About the >20 PINN-balls questions, I believe that demonstrating these effects could be valuable. And about the reply to question 8, I think the paper should include those results to support the statement in line 266 (the reader doesn't know whether they appeared nearly identical to the current ones).
> >
> > Again, thank you for the thorough replies! I will maintain my score for the reasons explained above, but I appreciate the rebuttal regardless!

---

> > > ### Author Response · Authors · 2025-08-05
> > >
> > > We thank the reviewer for their thoughtful follow-up and appreciate their continued engagement with our work. We are sorry to hear that our previous response did not fully resolve the reviewer’s concerns. Due to the constraints of the rebuttal format, some points were addressed in a necessarily concise manner. We are grateful for this opportunity to elaborate further.
> > >
> > > **Experimental clarifications and additional results**
> > > As indicated in our previous reply, we indeed plan to incorporate all clarifications related to experimental clarity into the revised manuscript. This includes addressing Question 8 by including the missing results to support the claim in line 266. We agree that these additions are important to ensure the paper is self-contained and reproducible.
> > >
> > > **On matrix-free solvers (e.g., Conjugate Gradient)**
> > > We now expand on our position regarding matrix-free approaches, particularly CG, and their limitations in the context of PINN Balls. We hope this deeper discussion clarifies our rationale for favoring sparse direct solvers over matrix-free methods in this setting.
> > >
> > > There are two primary challenges that hinder the practical application of CG for solving the Levenberg–Marquardt (LM) step in our framework:
> > >
> > > 1) _Convergence limitations due to ill-conditioning_: The LM step requires solving a global system that aggregates information across all subdomains. Matrix-free CG solves this indirectly, and its performance is highly sensitive to the spectral properties of the matrix. Unfortunately, as highlighted in [4, Figure 2b], this matrix often has a large condition number and a significant portion of its spectrum clustered near zero. As a result, convergence of CG may require many iterations and, without robust preconditioning, become prohibitively slow or even unstable. By contrast, sparse direct solvers solve the system exactly and are better suited to capturing the curvature information essential to second-order methods, especially in stiff or ill-conditioned regimes.
> > > Moreover, sparse direct solvers allow for the use of lightweight but effective preconditioning strategies, which are not easily applicable in matrix-free settings since the matrix is never explicitly formed.
> > >
> > > 2) _Practicality of Jacobian-vector products and GPU implementation_: While CG theoretically avoids full Jacobian construction, in practice, computing Jacobian-vector products via reverse-mode automatic differentiation is challenging to optimize efficiently on GPUs — especially without a structured matrix representation. In contrast, our approach constructs and operates on sparse matrices directly, leveraging structure and sparsity to keep computation tractable even as the number of model parameters increases. The modest overhead introduced during construction is offset by the efficiency of the sparse solver, which avoids the potential instability and parameter tuning challenges of iterative matrix-free approaches.
> > > This also connects with the reviewer's intuition in Question 9: while ill-conditioning may degrade the performance of CG, sparse direct solvers in our framework are robust to such issues and maintain stable performance across different PDE regimes.
> > >
> > >
> > > Finally, regarding our claim that "matrix-free methods could benefit from the sparse structure introduced by our method", we refer in particular to the suggestion of [25, Equation 10.41 and below] of solving the alternative LM step through the approximate matrix-free algorithm CG-Steihaug [25, Algorithm 7.2], which does indeed benefit from sparse construction. Without sparsity, computing Jacobian-vector products using reverse-mode automatic differentiation across all collocation points and PDE components quickly becomes intractable — especially on GPU, where memory bandwidth is limited and memory access patterns can significantly impact runtime. Hence, sparse direct solvers are still better suited in practice. While CG avoids explicit matrix inversion, it still needs to solve the LM step iteratively. However, the matrix involved is often ill-conditioned, and its spectrum typically includes many near-zero eigenvalues [4, Figure 2b]. This leads to slow convergence: a large number of iterations may often be needed to reach acceptable accuracy — making each LM step expensive. Sparse direct solvers, by contrast, solve the system exactly and preserve curvature information, which is especially beneficial in ill-conditioned regimes common in second-order optimization of PINNs. Moreover, they can benefit from simple preconditioners that are not available in matrix-free settings, since the matrix is never explicitly formed.
> > >
> > > In summary, while matrix-free methods remain a theoretically attractive option for memory-constrained regimes, our DD-based sparsity framework is essential to make them tractable, and in our current setting, sparse direct solvers are the more suitable and scalable choice. We will revise Section 4.1 accordingly to clearly express this nuanced perspective.

---

> > > > ### Comment · Reviewer_MZup · 2025-08-05
> > > >
> > > > Thanks for following up! Regarding the linear algebra, I appreciate the thorough explanation for why the submission chooses direct sparse solvers over matrix-free methods. However, even though I agree with a number of points in the reply above, I am not sure I agree with some of the critical ones (see below), which is why I maintain my original assessment.
> > > >
> > > > About the first point, I see that ill-conditioning could make conjugate gradients (or other Krylov methods, but in the rest of this text I'll only write "CG") problematic. However, whether or not this ill-conditioning is actually a critical factor in practice is unclear without trying it (ideally, in combination with an off-the-shelf preconditioner). My previous comment stated that I would need an explicit reason why CG can't be used or a benchmark that compares the solvers to change my score. In my view, the explanation in "1." above is not an explicit reason why CG can't be used, but it could be a starting point for a benchmark that shows the lack of performance of CG compared to direct sparse solvers. Still, as is, the reasoning in "1." did not change my mind even though I appreciate the explanation.
> > > >
> > > > About the second point, I am not sure I agree with this reasoning. There are many software implementations of efficient Jacobian vector products, e.g. _jax.jvp_ or _functorch.jvp_, which play well with GPUs in my experience (same for VJPs). If there is concrete evidence for the statement about the inefficiency of JVPs and VJPs on GPUs, that would be a valuable addition to the paper. As is, I am not entirely convinced; nonetheless, I am grateful for the explanation.
> > > >
> > > > In general, I still like the paper, and my original review already recommends accepting it. However, given that the CG-related linear-algebra gap persists, I maintain the borderline score.

---

> > > > > ### Author Response · Authors · 2025-08-06
> > > > >
> > > > > We thank the reviewer once again for the precious feedback, and will include for clarity the key points of our discussions in the Limitation section in a revised version of the manuscript.

---

### Official Review · Reviewer_MXFf · 2025-07-02

**Clarity:** 3
**Significance:** 3
**Originality:** 3
**Rating:** 5
**Confidence:** 4

**Summary:**

This paper introduces an ensemble model for PINNs, where each submodel learns a compactly supported Radial Basis Function (RBF). By ensuring the supports of RBFs cover the entire domain, the ensemble model autonomously learn domain decomposition. Employing the Levenberg-Marquardt update rule and Adversarial Adaptive Sampling (AAS) with a specified probability density function, it achieves notable performance. Furthermore, the paper includes theoretical extension of [33] to a more general class of regularizers.

**Questions:**

1. If there is an empirical result on complex domain such as dolphin-shaped interface, it would strengthen the generalizability of the proposed method.
2. The theorem said that probability density function $p_n$ converges to a uniform distribution. If this is observed in real experiments, could we expect the similar results by using uniform distribution rather than using AAS?
3. Considering the ensemble nature of the proposed framework, could the submodels be added in an adaptive manner? For example, after training PINNBalls-10, would it be possible to introduce an additional set of 10 submodels whose RBF centers are initialized in regions exhibiting high residual error? If the support of these new submodels is restricted to those high-error regions, this strategy might enable further improvement in accuracy without degrading performance in areas that are already well-learned.

**Ethical Concerns:**

["NO or VERY MINOR ethics concerns only"]

**Final Justification:**

Although my questions are not fully satisfied, they are not so significant to decrease ratings: Original paper include applications on non-trivial domain (simpler than dolphine, but sufficiently non-trivial) and applying adaptive strategy is potential benefit of the proposed method rather than necessary experiment to prove its performance. Consistency between theoretical and experimental result on trained distribution should be declared and rebuttal sufficiently has addressed it. I still suggest acceptance and maintain my rating.

**Limitations:**

yes.

**Paper Formatting Concerns:**

There are typos:
line 46: theses &rarr; these
line 287: in in &rarr; in
Figure2: BINNBalls 5 &rarr; PINNBalls 5
line 308: ouperform &rarr; outperform
line 340: stablish &rarr; establish

**Quality:**

3

**Strengths And Weaknesses:**

Strengths
1. The proposed method autonomously learns domain decomposition without requiring any a priori knowledge and eliminates the need to explicitly manage interfaces between subdomains.
2. The model outperforms existing baselines across several benchmarks.
3. The theoretical results on Adversarial Adaptive Sampling (AAS) are extended to more general settings, increasing their practical relevance.

Weaknesses
1. While accuracy improves with an increasing number of submodels, this comes at the cost of learning stability. Such instability may hinder deployment in real-world applications.
2. Although the concept of learnable domain decomposition is appealing, the paper lacks experimental validation on complex domains where such decomposition would be most advantageous.
3. Several important implementation details are provided only in the appendix and are not referenced in the main text. For example, the network architecture used in experiments is described in the appendix, but not mentioned or cited in the main body of the paper.

---

> ### Author Rebuttal · Authors · 2025-07-30
>
> We sincerely thank the reviewer for the positive feedback and for the insightful comments that help improve the clarity and rigor of our manuscript. Regarding the concern raised about training stability, we observe that fluctuations in the loss function primarily occur in low-loss regimes, typically when the training is nearing convergence. In practice, applying an exponential decay to the learning rate of AAS effectively mitigates this instability and stabilizes the training process. We will make this aspect more explicit in the final version of the paper and, in line with the reviewer’s suggestion, include additional implementation details in the main body of the manuscript.
>
> We now address the specific points raised:
> 1. While we did not explicitly consider cases involving highly irregular interfaces such as dolphin-shaped domains, the reviewer can refer to our Navier-Stokes experiments for a qualitative understanding of how the learned basis behaves in more complex scenarios. In such settings, the domain decomposition (DD) mechanism typically allocates higher resolution to regions where the PDE solution exhibits sharp gradients or localized complexity. This behavior is naturally guided by the residual-based adaptation mechanism embedded in the training process.
> 2.  We appreciate the reviewer’s insightful observation regarding the distribution of submodels. Indeed, our initialization procedure yields a probability density function that is approximately uniform. In relatively simple or linear scenarios—such as the Helmholtz equation—we observe that the domain decomposition remains mostly unchanged during training, which aligns with theoretical convergence expectations. However, in nonlinear settings, such as the Navier-Stokes equations, the AAS paradigm introduces meaningful deviations from the uniform distribution. These deviations serve to allocate more submodels in regions with higher solution complexity (e.g., steep gradients). While theoretical convergence to a uniform distribution assumes that the residuals vanish entirely — a reasonable but idealized assumption — this is rarely the case in practice. Consequently, the final distribution does not always converge to uniformity, but this deviation often enhances the model's accuracy. We view this tension between theory and practice as an important and interesting topic that merits deeper exploration in future work.
> 3. We find the reviewer’s proposed approach—dynamically adding submodels when convergence plateaus—both elegant and highly promising. This strategy could help avoid overparameterization by preventing the initialization of an unnecessarily large number of basis functions, and instead adaptively finding an optimal balance between submodel count and resolution. Furthermore, leveraging transfer learning for the initialization of newly introduced submodels could prevent disruption of ongoing training. We believe this direction holds great potential and plan to pursue it in future research.
>
> We are also grateful to the reviewer for pointing out typographical errors in the manuscript. These have been promptly corrected in the revised version.

---

> > ### Comment · Reviewer_MXFf · 2025-08-01
> >
> > Thanks for the responses. I read comments and the response to first and third points are suffice to me.
> > For the second point, I agree that the theoretical result often does not coincide with the practical observations, and deviation from initial or uniform distribution could be beneficial to attain better solution. However, I was confused and considered as if the theory explains how the distribution converges in practice. I would appreciate if the revised version clarify which aspect is beneficial in practice: theoretically guaranteed convergence or adaptive sampling far from uniform. It would be better if we can compare the result with that using uniform distribution, or a figure illustrate how the distribution changes during training.
> > Believing the discussion on weaknesses will be reflected as well, I will keep the current rating.

---

> > > ### Author Response · Authors · 2025-08-05
> > >
> > > We thank the reviewer for their response and are pleased to know that our clarifications were satisfactory. In the revised version of the manuscript, we plan to more clearly highlight the changes in the distribution of submodels.

---

### Official Review · Reviewer_DRVa · 2025-07-03

**Clarity:** 3
**Significance:** 3
**Originality:** 3
**Rating:** 3
**Confidence:** 4

**Summary:**

PINN Balls is a model that combine second-order training with domain decomposition on PINNs. An overlapping RBF-based domain decomposition ("balls”) is used to create a sparse Jacobian. Sparse Levenberg–Marquardt updates are adopted for second-order optimization. Adversarial Adaptive Sampling (AAS) is used to learn the decomposition along with the PINN training. Experimental results are presented on 2D Helmholtz, Burgers, and Navier–Stokes equations.

**Questions:**

1. Consider adding experiments on larger-scale PDEs, e.g., 3D or fine-resolution 2D, where dense LM cannot run due to GPU memory limits, but sparse LM succeeds and outperforms other models.

2. How sensitive is the method to the initialization of RBF centers and radii?

3. PINN Balls relies on AAS to train the DD. What are the benefits of AAS to residual-based methods? An ablation study of PINN Balls with residual-based sampling and AAS (with Dirichlet/entropic regularization) reporting final errors and convergence speed would be a good addition in the appendix.

**Ethical Concerns:**

["NO or VERY MINOR ethics concerns only"]

**Final Justification:**

I appreciate the authors response but will stick to my original score. For the canonical problem of flow past a cylinder, authors note that PINN (dense LM) runs on only a relatively small number of parameters (unclear how small?) and in this setting, the average velocity field is reasonably approximated but the unsteady dynamic of vortex shedding is not captured. However, there are no visualizations to validate these claims. In general, the scaling of PINN balls needs stronger validation.

**Limitations:**

yes

**Quality:**

3

**Strengths And Weaknesses:**

Strengths:

-- Scaling higher-order methods is an open problem. This paper considers this problem along with learning the adaptive domain decomposition.

-- Results on Helmholtz especially is impressive.

-- Extension of AAS theory, proving convergence to sampling under Dirichlet or entropic regularization.

Weaknesses:

-- Although sparse Jacobians allow for larger models, all benchmarks are on small 2D PDEs solvable by dense Levenberg–Marquardt. There is no demonstration of a problem that dense second-order methods OOM but PINN BALLS completes.
Moreover, assembling the CSR matrix has non-negligible overheads at moderate to large scale and could outweigh the gains from sparsity. GPU sparse kernels are also not as efficient as dense kernels.

-- Fluctuating losses with increasing balls point to unstable AAS maximization.

-- Benchmarks are three textbook PDEs on 2D domains. No demonstration on multiphysics and heterogeneous boundary conditions where PINNs often struggle.

-- Comparisons to XPINN/APINN use reported numbers rather than reproduced results under identical settings.

This paper presents good small-scale results, but falls short on demonstrating practical scalability which is the motivation for sparsifying second-order PINNs.

---

> ### Author Rebuttal · Authors · 2025-07-30
>
> We would like to express our sincere appreciation to the reviewer for highlighting several important areas for improvement in our work. We have carefully considered the points raised, and we address them in detail below.
> 1.  Domain decomposition enables tractability in large-scale or fine-resolution 3D problems through the parallel, block-wise construction of the Jacobian matrix. Importantly, the main computational bottleneck in the Levenberg-Marquardt (LM) optimization step — the inversion of the matrix $D_\theta L^T D_\theta L$ — depends solely on the number of trainable parameters in the model, which is the core focus of our paper. Notably, the time-dependent 2D Navier-Stokes example already illustrates a  scenario where applying a dense LM solver leads to out-of-memory errors. For quantitative context, in our GPU implementation, the LM update fails for architectures with more than approximately 2,000 parameters due to limitations in computing the dense Jacobian. While prior work has shown that second-order methods enable accurate training of PINNs even with a low number of parameters, the strategy we propose can extend this framework to a more consistent number of parameters, which is at times necessary to predict certain PDE solutions. For instance, in our experience, 2000 parameters are insufficient to model the vortex shedding behind a cylinder that we present. We intend to emphasize this point more clearly in a revised version of the manuscript.
> 2.  Under typical conditions, model performance is not highly sensitive to the initial number or placement of basis functions. However, careful initialization—as described in our paper—is essential for preserving the sparsity structure of the Jacobian $D_\theta L$. Excessively many or poorly placed basis functions can result in significant computational overhead when constructing the matrix $D_\theta L^T D_\theta L$, a limitation we have discussed in the paper but will clarify further.
> 3.  To the best of our understanding, the AAS mechanism functions as a residual-based method. It adaptively concentrates model capacity on regions with large residuals, thus facilitating a more effective domain decomposition. In addition to its conceptual alignment with DD, AAS also offers desirable convergence behavior. Empirically, we find that AAS improves model accuracy by one to two orders of magnitude in certain nonlinear regimes, such as near the cylinder in the Navier-Stokes setup. We will make this mechanism and its empirical benefits more explicit in the revised version of the paper.
>
> While we did not directly reproduce the results of XPINN and APINN, we aimed to ensure a meaningful and fair comparison by adapting our model to align closely with their training setups — for instance, by matching the number of collocation points, model parameters, and activation functions wherever applicable.
> In addition, the reviewer raises a valid concern regarding the oscillatory behavior of the loss function when PINN Balls are instantiated with multiple submodels. We acknowledge that these fluctuations often stem from instability in the AAS maximization. However, in our experiments, such behavior typically occurs only in very low-loss regimes, where the optimization is approaching convergence. Employing a stronger exponential decay in the AAS update tends to mitigate these fluctuations effectively. We will ensure these points are better articulated in the manuscript.

---

> > ### Comment · Reviewer_DRVa · 2025-08-05
> >
> > Thanks for the responses. Regarding dense LM PINN leading to out-of-memory errors for 2D Navier-Stokes, Fig 2 (right) shows Navier-Stokes PINN with dense LM converging to a loss ~10^-3. Can you clarify what you're referring to in the paper?

---

> > > ### Author Response · Authors · 2025-08-05
> > >
> > > We thank the reviewer for engaging in discussion. The loss curve in Figure 2 (right) shows the training loss across iterations for PINN Balls applied to the Navier-Stokes equations. The “LM PINN” curve corresponds to a baseline model trained using a dense LM step that fits in GPU memory — i.e., one constrained to a relatively small number of parameters. Regarding the observed convergence to a loss of approximately $10^{-3}$ , we note that for the canonical problem of flow past a cylinder, such a loss typically corresponds to a steady-state solution. In this regime, the average velocity field is reasonably approximated, but the unsteady dynamic of vortex shedding is not captured. In contrast, lower training loss values, as achieved by the PINN Balls 20 model, are indicative of models that capture these finer-scale, time-dependent behaviors more faithfully. The results for this model is included in the appendix. We hope this clarifies the distinction and addresses the reviewer’s concern. We remain available for further questions or discussion.

---

### Official Review · Reviewer_gnjE · 2025-07-03

**Clarity:** 2
**Significance:** 3
**Originality:** 3
**Rating:** 4
**Confidence:** 4

**Summary:**

The paper introduces PINN BALLS, a sparse Mixture-of-Experts architecture for Physics-Informed Neural Networks (PINNs).
Key ideas are:
- trainable radial-basis “balls” that decompose the domain and create a partition of unity.
- second-order Levenberg-Marquardt updates applied locally so Jacobians stay sparse and fit GPU memory.
- Adversarial Adaptive Sampling (AAS) used as the gating distribution, with new convergence theorems that replace the original Lipschitz constraint by Dirichlet, and entropy-based regularizers.
- extensive empirical study on Helmholtz, Burgers, and 2-D Navier–Stokes benchmarks plus ablations on memory/runtime.
A comparison table highlights how the approach differs from XPINN, FBPINN and APINN (sparsity, no interface loss, adaptive sampling, second-order training)

**Questions:**

- Have you attempted to scale PINN BALLS to 3D problems or complex multi-physics PDEs? Would sparse Jacobian construction and solver performance remain tractable in such high-dimensional or coupled systems?
- Could you elaborate on why you chose quartic radial basis functions for the gating balls? Did you explore other compactly supported kernels or make the RBF shapes learnable? How sensitive is the model to this design choice?
- How sensitive is model performance to the number or initial placement of balls?
- Would your method require significant architectural or training changes to handle time-dependent PDEs.
- Can you provide more quantitative analysis of the time and memory trade-offs of sparse Jacobian construction and sparse LM solving compared to dense LM or first-order optimizers?

**Ethical Concerns:**

["NO or VERY MINOR ethics concerns only"]

**Limitations:**

- Reproducibility is limited by the lack of a code release at submission time, though the authors state their intent to share it. Open access would allow others to test and verify scalability claims.

**Paper Formatting Concerns:**

No concern

**Quality:**

2

**Strengths And Weaknesses:**

Strengths:
- While mixture-of-experts in PINNs is not a new idea, and prior works already explored sparse gating and soft domain decomposition, the idea proposed here is to combine the benefits of DD, second-order training methods and Adversarial Adaptive Sampling (AAS)
- A radial basis “ball” formulation is derived for soft domain decomposition that leads to both partition-of-unity gating and Jacobian sparsity.
- The paper provides theoretical convergence results for Adversarial Adaptive Sampling (AAS), relaxing prior assumptions and generalizing earlier proofs using Dirichlet and entropy regularizers.
- In addition to empirical evaluation, Authors provide ablation studies across ball count, sampling, and sparsity.
- Transparent discussion of limitations such as sparse Jacobian overhead and current AD support issues, which raises my respect and trust to their science work quality.

Weaknesses:
- No ablation study on the effect of different components the authors proposed in their study. For example, comparing the following variations can shed light on which components are truly necessary and and where most of the gains come from: Full PINN BALLS, vs without AAS, vs Without LM update step, etc.
- Lack of comparison with non-PINN state-of-the-art models: Since the ultimate goal is to train an accurate surrogate for PDE solutions, it is important to evaluate how PINN BALLS performs against other state-of-the-art models in terms of predictive accuracy, regardless of the underlying optimization algorithm or architecture type. I suggest that the authors include a comparison with data-driven models (such as FNO or DeepONet) as part of their experimental evaluation.
- No public code at the time of submission exists and their answer to NeurIPS checklist question must be "No". This can limit the reproducibility. Even an anonymized repo for the results reported in this paper would help.
- The notation can be dense and intimidating for readers unfamiliar with second-order methods and sparse linear algebra. A high-level algorithm box or diagram of the training pipeline would improve clarity.

---

> ### Author Rebuttal · Authors · 2025-07-30
>
> We sincerely thank the reviewer for the positive feedback and for the constructive comments that help us improve the quality of our manuscript. In particular, we acknowledge the lack of an ablation study analyzing the contributions of the algorithmic components, specifically the AAS and the LM optimization step. From our experience, the LM step plays a critical role in achieving the reported results. In the context of domain decomposition, first-order optimization methods tend to show a significantly slower convergence as the number of basis functions increases. This is especially problematic in nonlinear regimes such as those arising in the Navier-Stokes equations, where convergence is often not achieved. Conversely, the AAS component can improve model accuracy by one to two orders of magnitude in such nonlinear settings, although its impact is minimal in linear cases. We plan to explicitly incorporate this discussion, along with a high-level diagram of the algorithm - as suggested - , in a revised version of the paper.
>
> Below, we provide specific responses to the reviewers’ questions:
> * The central focus of our work is to demonstrate scalability with respect to the number of trainable parameters — arguably one of the primary bottlenecks in applying second-order optimization to overparameterized physics-informed models. While we do not directly address time-dependent 3D problems or complex multiphysics scenarios in this study, the proposed methodology is designed keeping such extension in mind. In fact, high-dimensional and multiphysics problems often demand the precision and convergence speed of second-order methods to be tractable. Even in the time-dependent 2D setting of the  Navier-Stokes equations considered here, we encounter severe memory limitations: the computation of the dense Jacobian $D_\theta L$ and its inversion leads to an out-of-memory error at approximately 2000 parameters on our GPU implementation, due to the need to backpropagate through all collocation points and equations of the PDE. Our domain decomposition strategy directly addresses this scalability issue by enabling a block-wise construction of the matrix $D_\theta L^T D_\theta L$. Notably, the dimensionality of $D_\theta L^T D_\theta L$ depends only on the number of model parameters, which means that at this point in the algorithm memory growth is decoupled from domain/PDE complexity. This allows our framework to remain tractable in principle even as the problem dimensionality or physical complexity increases — a direction we view as a natural extension of this work.
> * The selection of RBFs in our framework is relatively flexible. Our aim was to ensure compact support and avoid steep derivatives. We experimented with Gaussian kernels (non-compact support) and second-order finite-element bases (with discontinuous first derivatives) and observed no significant impact on model performance. This robustness can be attributed to the efficacy of second-order optimization in the training process.
> * Under reasonable conditions, the model’s performance is not highly sensitive to the initial number or placement of basis functions. Nevertheless, a well-chosen initialization—such as the strategy proposed in our paper—is important to maintain the sparsity of the Jacobian $D_\theta L$. Excessive or poorly placed basis functions can lead to substantial computational overhead during the construction of $D_\theta L^T D_\theta L$, as discussed in the Limitations section.
> * This is an excellent point. In principle, domain decomposition in PINN Balls is not restricted to spatial variables. In the presented Navier-Stokes experiment, we address a time-dependent case, demonstrating that the approach is compatible with temporal domains. One could implement DD either in space alone or across both space and time, depending on the application. We believe that future work could significantly benefit from more sophisticated time-marching strategies, and we intend to explore this in subsequent research.
> * The Jacobian is constructed by iterating over each submodel, requiring backpropagation to compute partial derivatives for each. This cost is roughly proportional to computing a standard physics-informed loss, multiplied by the number of submodels. For a standard implementation that loops over the model, this means that the computational time of each step would grow linearly with the number of submodels. Therefore, adopting an efficient and parallel implementation is essential to keep this step tractable and the overhead minimal. Regarding solver performance, Figure 1 illustrates the advantage of a sparse LM solver compared to its dense counterpart. In principle, a dense solver is more effective, however, the dense implementation fails before reaching 8000 parameters, in contrast to the sparse model which deals with more than 10'000 with ease. When contrasted with first-order methods, the LM step achieves several orders of magnitude improvement in loss with a fraction of the iterations (few thousands vs. several hundreds of thousands). Moreover, in the DD setting—especially for complex problems like the Navier-Stokes equations first-order methods fail to converge altogether. We will include this discussion, in a revised version of the paper.
>
> Lastly, we acknowledge the reviewer’s note regarding code availability. The release of the code is currently under discussion with our primary partner institution. However, we are happy to share it upon request through the corresponding author.

---

### Note · Authors · 2025-08-15

We would like to sincerely thank the reviewers for their invaluable feedback and thoughtful comments. The discussion has been extremely helpful in identifying areas where our paper can be improved in clarity, completeness, and accessibility. We will use the insights gained from the review process to substantially strengthen the revised manuscript. We are committed to improving the paper so that it is not only technically rigorous but also clear to the reader. In particular, we plan to:
- Include a high-level diagram of the algorithm to better illustrate each training step and component to the reader.
- Highlight the performance differences between first-order and second-order optimization in our DD setting.
- Address the option of matrix-free solvers (as CG) and their interaction with our DD setting, as an alternative to the application of a sparse direct solver.
- Provide a detailed discussion of the contribution and effect of each algorithmic component.
- Clarify the impact of different initialization strategies for the domain decomposition.
- More thoroughly address the out-of-memory issues of plain second-order methods and how they already limit the application to the Navier–Stokes use case.
- Correct typographical errors and expand on practical implementation details, as well as address all other points raised in the discussions with the reviewers.

While we were unfortunately not able to engage in a thorough discussion with all reviewers, we hope our rebuttal addressed their main concerns satisfactorily.

We believe this work makes a meaningful contribution by addressing a key bottleneck in the adoption of second-order methods for training PINNs - the scalability in the parameter size - by leveraging DD and AAS. Furthermore, we provide novel theoretical background on adaptive sampling for PINNs, which smoothly links to the problem of learning the DD alongside the model training. Taken together, these practices result in a methodology that is both practically effective and theoretically grounded, making it relevant to a broad range of problems in the field of scientific machine learning.

---

### Decision · Program_Chairs · 2025-09-17

**Decision:**

Accept (poster)

**Comment:**

(1) Summary of this work: This paper introduces PINN BALLS as a sparse MoE architecture for Physics-Informed Neural Networks (PINNs), which can effectively unite the domain decomposition, second-order optimization and adversarial adaptive sampling. The authors provide extensive experiments on different PDEs to verify the effectiveness of PINN BALLS.

(2) Strengths and weaknesses: One of the main strengths of this work is its seamless and sound unification of several techniques. Especially, the idea of scaling higher-order methods through learning the adaptive domain decomposition is novel, which is also acknowledged by several reviewers. An in-depth discussion of theoretical property is also included, along with a transparent description of limitations, showing favorable scientific rigor. However, one main concern of this paper is the term "scaling". Although the authors have provided experiments about "scalability" in model parameters, I think the "scaling" term is more about generalizability, namely, one model to different tasks, which is really hard to achieve in PINNs. Besides, the evaluation of scaling significance is also mentioned by Reviewer DRVa.

(3) Summary of rebuttal: Although Reviewer DRVa expresses his/her slight concerns about this paper's evaluation part, other reviewers think it should not be a reason for rejection, which I also agree with. Also, during the rebuttal, the authors have provided detailed information about the effect of each algorithmic component and clarification of experiments (They need to add this into the final version of this paper).

**Final decision:** After a comprehensive discussion with reviewers and carefully reading the rebuttal, I think the proposed method can be inspiring for future applications of PINN in handling complex problems, and it can tackle the bottleneck in the adoption of second-order methods for training PINN. Thus, I would recommend acceptance, but not a higher score, due to the limitation in evaluation. And I strongly suggest the authors add more baseline methods and a more comprehensive evaluation of the scaling property in the final paper.